# Genetically regulated eRNA expression predicts chromatin contact frequency and reveals genetic mechanisms at GWAS loci

Michael J. Betti [1] ✉, Phillip Lin[1], Melinda C. Aldrich [1] & Eric R. Gamazon[1,2] ✉

The biological functions of extragenic enhancer RNAs and their impact on disease risk remain relatively underexplored. In this work, we develop in silico models of genetically regulated expression of enhancer RNAs across 49 cell and tissue types, characterizing their degree of genetic control. Leveraging the estimated genetically regulated expression for enhancer RNAs and canonical genes in a large-scale DNA biobank (N > 70,000) and high-resolution Hi-C contact data, we train a deep learning-based model of pairwise three-dimensional chromatin contact frequency for enhancer-enhancer and enhancer-gene pairs in cerebellum and whole blood. Notably, the use of genetically regulated expression of enhancer RNAs provides substantial tissue-specific predictive power, supporting a role for these transcripts in modulating spatial chromatin organization. We identify schizophrenia-associated enhancer RNAs independent of GWAS loci using enhancer RNA-based TWAS and determine the causal effects of these enhancer RNAs using Mendelian randomization. Using enhancer RNA-based TWAS, we generate a comprehensive resource of tissue-specific enhancer associations with complex traits in the UK Biobank. Finally, we show that a substantially greater proportion (63%) of GWAS associations colocalize with causal regulatory variation when enhancer RNAs are included.

Enhancers are essential mediators of gene expression, regulating spatial and temporal expression patterns through recruitment of DNA-binding proteins and establishment of chromatin conformation[1]. While the question of how enhancers mediate chromatin activity and gene expression has been well-studied, the biological functions of enhancer RNAs (eRNAs), the RNAs transcribed from these regulatory elements, remain relatively under-explored[2]. Research over the past decade has suggested that eRNA transcription plays a key role in mediating gene transcription[3], facilitating chromatin modifications and enhancer loop formation[4,5], and driving cell fate determination[2]. While canonical genes – mRNAs and long non-coding RNAs (lncRNAs) – are usually spliced and polyadenylated and transcribed from promoters, eRNAs exhibit a higher degree of structural and functional diversity and can

be sub-divided into two distinct categories[6]. The first sub-group, 1D eRNAs, is composed of long, spliced, polyadenylated, unidirectionally transcribed transcripts that can function in *trans*[4]. The second sub-class, 2D eRNAs, is composed of short, unspliced, non-polyadenylated, bidirectionally transcribed transcripts that function in *cis*[7], and most eRNAs fall into this latter category. As canonical genes are influenced by expression quantitative trait loci (eQTLs), we expect that eRNA expression should likewise be under at least some degree of genetic control. Additionally, because eRNAs play putative regulatory roles in a range of biological processes, it is reasonable to hypothesize that eRNA expression influences human complex traits, including disease risk.

Recent work investigating the role of eRNAs in neuropsychiatric traits strongly supports this hypothesis[8]. In one such study, the authors

[1]Department of Medicine, Division of Genetic Medicine, Vanderbilt University Medical Center, 2525 West End Avenue, Suite 700, Nashville, TN 37203, USA. [2]Clare Hall, University of Cambridge, Herschel Rd, Cambridge CB3 9AL, UK. ✉e-mail: michael.j.betti@vanderbilt.edu; eric.gamazon@vumc.org

mapped enhancer eQTLs, which they termed EeQTLs, in regions of the brain. They concluded that eRNA expression explains a substantial proportion of neuropsychiatric trait heritability (6.8%). Interestingly, these authors reported that the proportion of heritability explained by eRNAs is largely complementary to, rather than overlapping with, the proportion that can be explained by canonical gene expression alone.

In this work, we present predictive models of eRNA expression trained using whole genome sequencing (WGS) and eRNA expression profiled across 49 cell and tissue types[9] (Fig. 1a), quantifying its level of genetic control. Using a deep learning-based framework, we show that pairwise genetically regulated expression (GReX) predicts the three-dimensional chromatin contact frequency of an enhancer-gene or enhancer-enhancer pair (Fig. 1b), lending support to a modulating role in 3D spatial chromatin organization. We then perform an eRNA-based TWAS of schizophrenia (SCZ) risk (Fig. 1c), identifying eRNA associations that are independent of canonical genes, whose causality we investigate using Mendelian randomization (Fig. 1d). Finally, we apply this eRNA-based TWAS methodology across the UK Biobank[10,11] to

generate a comprehensive reference resource of tissue-specific enhancer associations with the human phenome (Fig. 1e).

## Results

### Genetic models of eRNA expression

We trained models[12–14] of genetically regulated eRNA expression, encompassing a total of 14,471 transcribed enhancers profiled across 49 human cell and tissue types. Because eRNAs were quantified from RNA-seq data, which underwent selection for polyadenylated transcripts prior to sequencing[9], 2D eRNAs, which are not polyadenylated, might be underrepresented in the training dataset. Previous characterizations of eRNAs have described 2D transcripts as having a length below 2 kb, while 1D eRNAs are longer than 2 kb[15–17]. Using transcript length, we characterized the proportion of eRNAs in our training dataset that fit the profile of a 2D eRNA versus a 1D eRNA. Of the 14,471 unique eRNA transcripts included in our trained models, 13,580 (93.84%) were less than 2 kb in length, while only 891 (6.16%) has a length greater than or equal to 2 kb (Supplementary Data 1). The

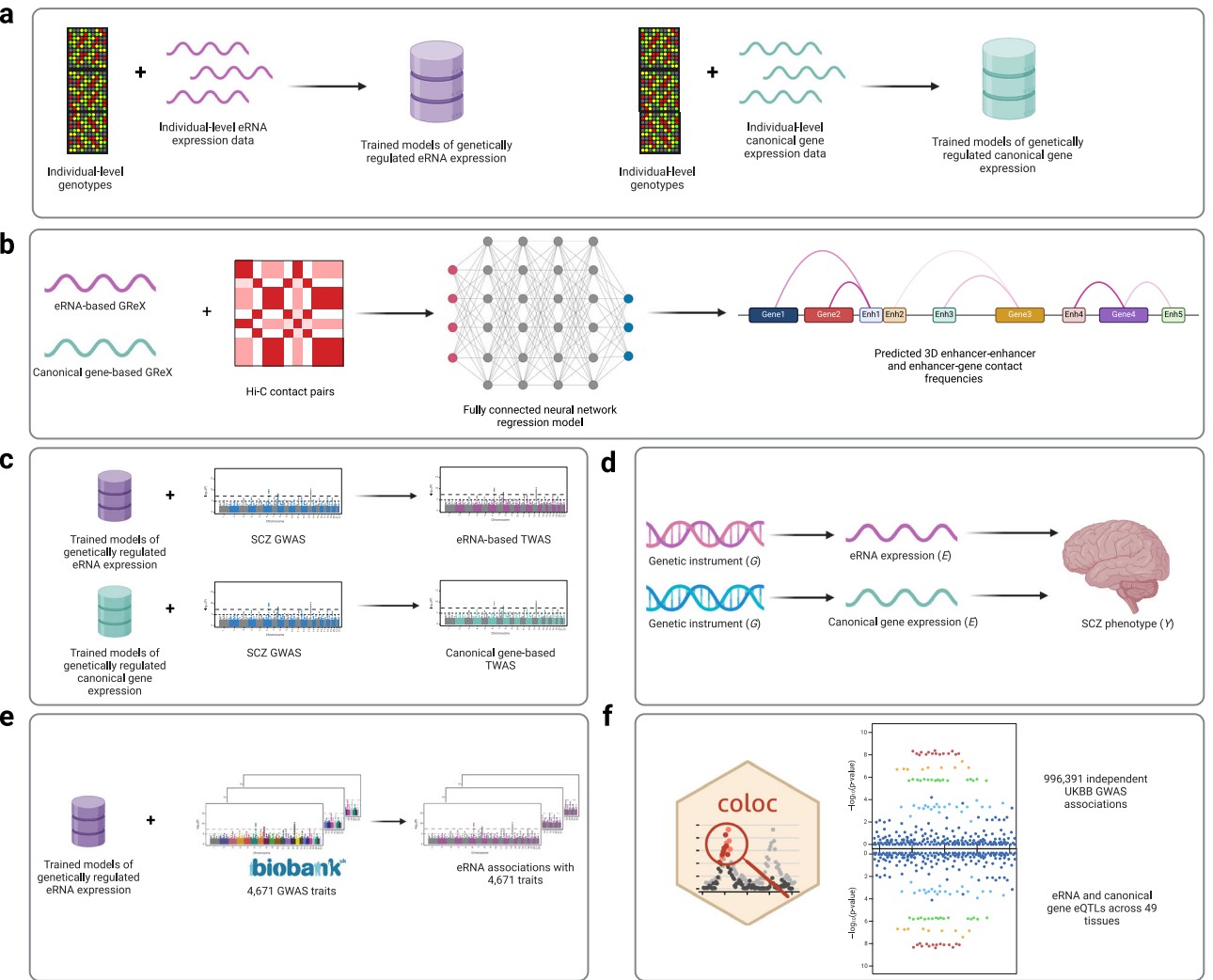

**Fig. 1 | Study workflow. a** In silico genetic models of genetically regulated expression (GReX) of eRNAs and canonical genes were trained using whole genome sequencing data and corresponding expression profiling generated by the GTEx Consortium. **b** Using the trained models, GReX of eRNAs and canonical genes was imputed across 72,828 BioVU samples. Mean predicted expression values were used to train deep learning models of three-dimensional contact frequencies observed in Hi-C contact matrices. **c** eRNA-based and canonical-gene-based TWAS of schizophrenia (SCZ) were run. **d** Genome-wide significant TWAS associations were tested for causality using Mendelian randomization. This allowed us to

identify loci with a causal SCZ-associated eRNA, gene, or both and explore the potential underlying mechanisms by which they could influence disease risk. **e** The eRNA-based TWAS models were applied the UK Biobank, generating a comprehensive resource of tissue-specific eRNA associations on a phenome-wide scale. **f** eQTL mapping and colocalization analysis was performed across nearly 1 million independent GWAS associations in the UK Biobank to identify the proportion of signals explained by an eRNA eQTL versus a canonical gene eQTL. All panels were created in BioRender[117].

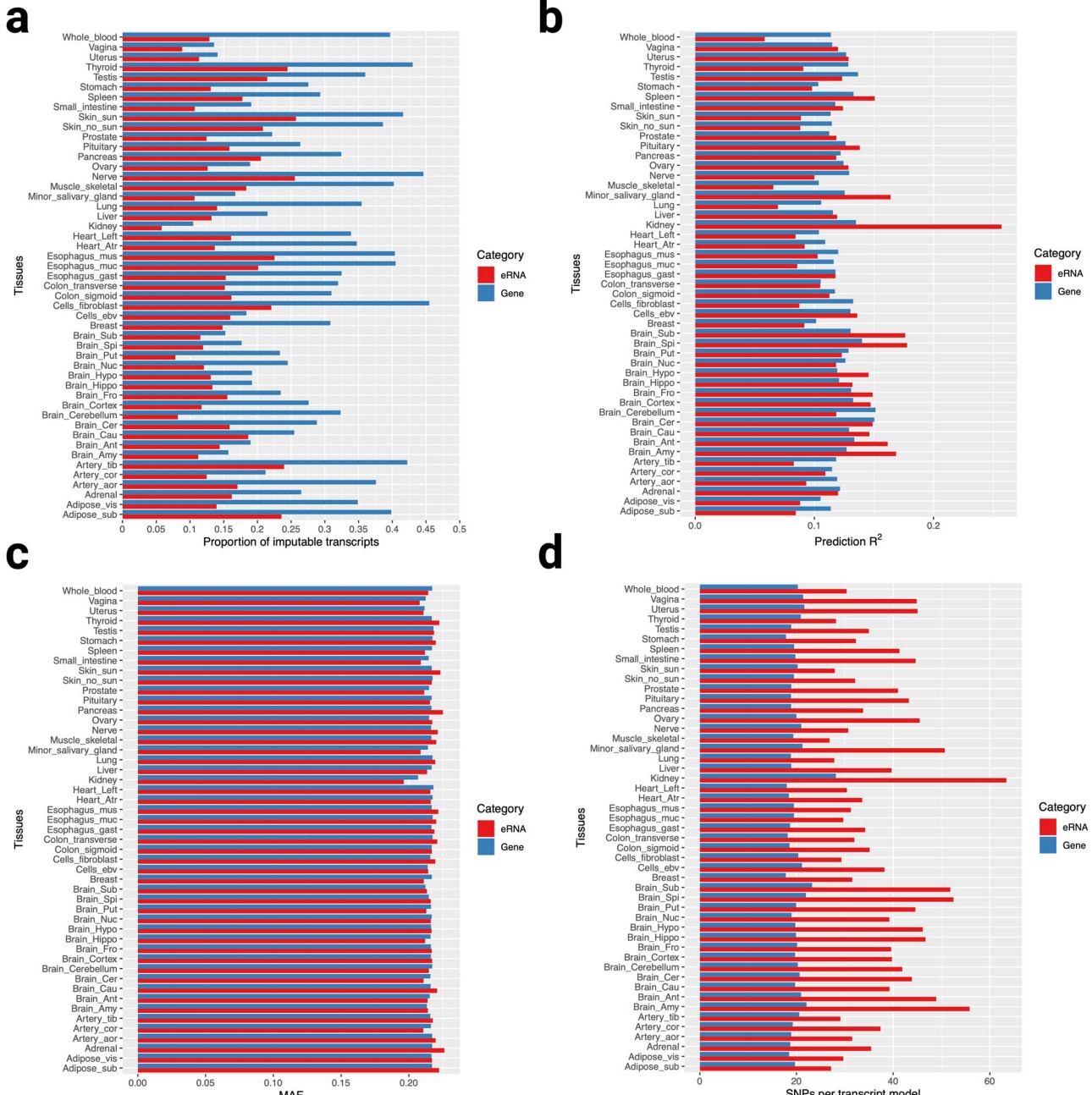

**Fig. 2 | Comparison of eRNA-based models with canonical-gene-based models in 49 GTEx tissues. a** Across all tissues, the eRNA-based models included a substantially lower proportion of imputable transcripts than the canonical gene-based models. **b** Mean prediction $R^2$ varied by tissue type, with some tissue models having a higher predictive performance for eRNAs and others having a higher performance for canonical genes. **c** The mean MAF of variants included in the models did not vary between eRNAs and canonical genes. **d** The eRNA-based models, across all tissues, yielded a higher mean number of SNPs per gene transcript than the canonical-gene-based models. Source data are provided as a Source Data file.

median eRNA transcript length was 550 bp (Supplementary Fig. 1). These results suggest that despite transcripts in the training set undergoing selection for polyadenylated transcripts, the vast majority still exhibit characteristics typical of 2D eRNAs.

We compared the resulting eRNA-based models with an analogous set trained on canonical gene expression. We evaluated the proportion of imputable transcripts, mean prediction $R^2$, mean minor allele frequency (MAF) of the SNP features, and mean SNPs/transcript ratio. For each tissue, we observed a higher proportion of imputable canonical genes than imputable eRNAs (Fig. 2a). Prediction $R^2$, however, showed greater cross-tissue variability (Fig. 2b). Across the 49 tissues, 21 had a higher mean prediction $R^2$ for the eRNAs than the

canonical genes. And while the mean MAF of model SNP features was essentially the same across all eRNA and canonical-gene-based models (Fig. 2c), we found that across all cell and tissue types, eRNA-based models had a significantly higher SNP per transcript ratio (mean 38.16 SNPs/gene versus 19.94 SNPs/canonical gene across all tissue models, $p < 2.2 \times 10^{-16}$) (Fig. 2d).

## Regression modeling of GReX as a predictor of contact frequency

Since enhancer-promoter interactions are among the most critical mechanisms of gene regulation, prediction of the functional consequences of genetic variation on chromatin contact could illuminate

disease mechanisms. We sought to assess the extent to which genetically regulated eRNA and canonical gene expression predict three-dimensional chromatin contact frequency.

We applied the eRNA and canonical gene models to BioVU[18] (N = 72,828), Vanderbilt University Medical Center's DNA biobank linked to electronic health records, to impute sample-level GReX. We then leveraged two high-resolution Hi-C datasets (4D Nucleome[19] Data Portal; Methods) that had been generated from the K562 leukemic cell line and primary astrocytes of the cerebellum. Within the K562 dataset, we identified 85,630 contact pairs that overlapped with enhancer-gene pairs with imputable expression (for each member transcript of the pair). Within the cerebellum dataset, we identified 95,701 contact pairs for enhancer-gene pairs with corresponding imputable expression. A training set was compiled for each tissue, consisting of the mean GReX values (estimated in BioVU) for each transcript pair, along with the normalized Hi-C contact frequency for the corresponding contact pair (Supplementary Figs. 2 and 3).

As a baseline, we fit a linear regression model to predict 3D contact frequency for enhancer-enhancer or enhancer-gene pairs using the mean predicted GReX of the corresponding transcripts in BioVU. An 80/20 train-test split was used to train the models. In whole blood, the training data consisted of 32,133 enhancer-gene pairs (where the eRNA was upstream and the gene was downstream), 34,682 gene-enhancer pairs (with the gene being upstream and the enhancer downstream), and 1689 enhancer-enhancer pairs. In the cerebellum training data, these numbers were 37,710; 36,117; and 2733, respectively. The resulting linear model for each tissue exhibited poor performance ($R^2 \approx 0$).

Due to the low performance of the initial linear model, we next trained four non-linear models (polynomial regression, random forest regression, support vector regression, and gradient boosting regression) to predict contact frequency using GReX data (Supplementary Figs. 4 and 5). Of the non-linear models, gradient boosting regression showed the highest performance in both whole blood ($R^2 = 0.08$ using 290 boosting stages) and cerebellum ($R^2 = 0.13$ using 290 boosting stages). The still-low performance justified the use of a more complex neural network-based approach.

### Training a baseline contact model using directly assayed expression

Prior to training a neural network using GReX data, we trained a baseline contact prediction model using eRNA and gene transcript counts quantified via a nuclear run-on assay in the K562 cell line (Supplementary Fig. 6). As nuclear run-on assays are considered a gold standard for detecting nascent eRNA transcription, a model trained on these expression data should be an ideal benchmark against which a GReX-based model can be compared.

An initial neural network model was trained using eRNA and canonical gene transcription to predict chromatin contacts in the K562 cell line. Hyperparameter tuning was conducted across a pre-defined search space, and five-fold cross validation was used to quantify model performance (see Methods). The optimal model architecture (Supplementary Fig. 7a) consisted of two neurons in the input layer (for the normalized expression levels of the upstream and downstream transcripts), two hidden layers with 150 neurons each (using a ReLU activation function), and a single output neuron. Hidden weights were initialized using a normal distribution, while those in the output neuron were initialized with zeros. The model was trained over 90 epochs using a batch size of 160. The Adagrad[20] optimizer was utilized with a learning rate of 0.2. This optimal model architecture achieved a mean $R^2$ of 0.23 across the validation folds (Supplementary Fig. 7b) and 0.27 in the independent test set.

### GReX of enhancer-gene pairs predicts chromatin contact frequency

We trained fully-connected neural networks to model the non-linear relationship between GReX and chromatin contact frequency in whole blood and cerebellum. The optimal model architecture for the whole blood-based model (Supplementary Fig. 8a) had two neurons in the input layer (for the GReX values of the upstream and downstream transcripts), two hidden layers with 120 neurons each (utilizing a hard sigmoid activation function), and a single output neuron. Hidden weights were initialized using a Kaiming uniform distribution, while those in the output neuron were initialized using a Kaiming normal distribution. The model was trained over 90 epochs with a batch size of 110, using the NAdam[21] optimizer with a learning rate of 0.01. This optimal model achieved a mean $R^2$ of 0.22 in both the validation folds (Supplementary Fig. 8b) and independent test set (Supplementary Fig. 8c). Notably, we found that the GReX-based model trained in whole blood maintained its predictive performance in the K562-derived nuclear run-on dataset ($R^2 = 0.15$).

Among the 17,126 transcript pairs in the test set, we observed a median relative error $E$ (see Methods) of 2.80 (Supplementary Data 2). The best-predicted contact pair in these data was between the enhancer ENSR00000041089 and gene *PELI3* (true value = 6.0, prediction = 5.95, $E = 7.0 \times 10^{-3}$), a protein-coding gene associated with erythrocyte and lymphocyte counts[22,23].

The optimal cerebellum-based model architecture (Fig. 3a) consisted of two hidden layers with 90 neurons each (using the Softsign activation function). Weights in these hidden layers were initialized following a Xavier normal distribution, while those in the output layer a uniform distribution. The model was trained for 50 epochs with a batch size of 80, using Adagrad[20] as the optimizer and a learning rate of 0.3. The optimal model achieved a mean $R^2$ of 0.37 across the validation folds and 0.38 in the independent test set (Fig. 3b), capturing non-linear patterns of GReX associated with contact frequency (Fig. 3c). Interestingly, the two-dimensional GReX profile for contact pairs was found to be constrained to a subregion of the total space (Fig. 3c).

In addition to the enhanced prediction $R^2$ in cerebellum, we observed a decreased median relative error in the cerebellum test set compared with whole blood ($E = 0.85$). The best-predicted contact pair among the 19,141 in these data was between enhancer ENSR00000186261 and *WDR41* (true value = 1.0, prediction = 0.99, $E = 1.80 \times 10^{-4}$), a protein-coding gene implicated in fronto-temporal dementia and amyotrophic lateral sclerosis[24,25] (Supplementary Data 3).

Next, using the same respective test sets for whole blood and cerebellum, we utilized SHapley Additive exPlanations (SHAP)[26] to determine the relative contributions of the upstream and downstream GReX features to model prediction. In both tissues, we found the mean relative contribution of the downstream transcript to be greater than that of the upstream transcript (60.67% versus 39.33% in whole blood and 53.06% versus 46.94% in cerebellum, Fig. 3d).

The 3D contact frequency model in each tissue converged with different optimal hyperparameters. We thus evaluated the cross-tissue portability. We postulated that a predictive model that can achieve good performance in a tissue not available during training may learn high-level principles underlying the global relationship between expression and chromatin contact, rather than more stochastic patterns specific to the training tissue.

We utilized the whole blood-based model to predict contact frequency across the previously unseen cerebellum data. Across the 95,701 unique enhancer-enhancer and enhancer-gene pairs in these cerebellum data, the model achieved a prediction $R^2$ of only 0.01, a marked decrease from the $R^2$ of 0.22 in the tissue-matched test set.

We also tested the performance of the cerebellum-trained model on the combined 85,630 transcript pairs in the whole blood data. The cerebellum-based neural network achieved markedly higher cross-tissue performance, with a prediction $R^2$ of 0.18 in whole blood, nearly matching the performance of the whole blood-based model itself in this tissue.

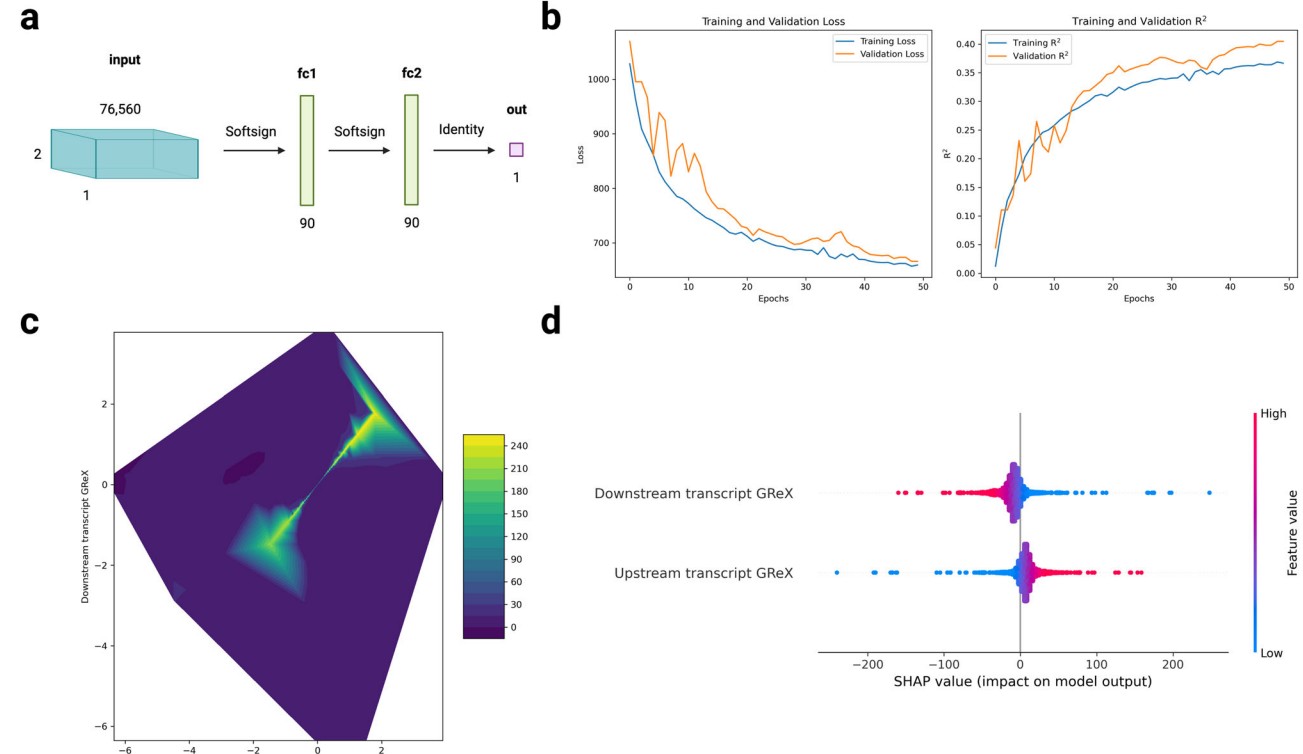

**Fig. 3 | Deep learning model with tissue-specific eRNA and canonical gene GReX as features predicts three-dimensional contact frequency. a** Grid search across 13 hyperparameters was used to find the optimal model architecture (see Methods). **b** The neural network was trained in cerebellum GReX and contact frequency data for 50 epochs, achieving a mean prediction R² of 0.37 across the validation folds and 0.38 in the independent test set. Within a second tissue type not previously seen by the model, whole blood, we observed a prediction R² of 0.18, which shows some cross-tissue portability. **c** Contact frequency prediction in the cerebellum test set (denoted by color) as a function of the GReX of the upstream and downstream transcript levels (x and y axes, respectively). The two-dimensional GReX space for contact pairs is constrained to lie in the colored region. **d** SHAP values representing the relative mean contributions of the upstream and downstream transcripts to contact frequency predictions, which we found to be 46.94% and 53.06%, respectively. Source data are provided as a Source Data file.

## Contact frequency shows low negative correlation with genomic distance

Previous work has demonstrated that linear genomic distance between two loci is generally a predictor of their contact frequency[27]. However, unlike (tissue type independent) genomic distance, 3D chromatin interaction exhibits highly tissue-specific patterns, especially involving a non-ubiquitously expressed gene[28].

To explore the degree to which genomic distance might be influencing our predictions, we calculated the Pearson correlation of the association of the genomic distance for a transcript pair (used in the deep learning models' test sets) with both the predicted and observed contact frequency for the pair (Supplementary Fig. 9). Across the 17,126 transcript pairs in whole blood and across the 19,141 in cerebellum, we found the correlation with both predicted ($R = -0.05$ and $R = -0.08$, respectively) and observed contact frequency ($R = -0.10$ in whole blood and $R = -0.11$ in cerebellum) to be low, suggesting that the deep learning-based contact frequency predictions were largely independent of genomic distance.

## TWAS identifies eRNA and canonical gene associations with SCZ risk

A previous study[8] trained FUSION-based[29] models of enhancer expression in two brain regions, the dorsolateral prefrontal cortex (DLPFC) and anterior cingulate cortex (ACC). Utilizing these models, which included 8,702 unique enhancers, the authors performed a TWAS of SCZ[30], yielding 98 enhancer associations outside of the major histocompatibility complex (MHC) region. Using our eRNA models encompassing a total of 14,471 unique enhancers expressed across 49 cell and tissue types, including a more diverse sampling of brain

regions, we performed TWAS using the same SCZ GWAS results. Despite a higher multiple testing threshold for genome-wide significance ($p < 1.23 \times 10^{-6}$), we identified 392 significant enhancer-tissue associations outside of the MHC region (Fig. 4a, b), including 114 in the brain (Fig. 4c, d), representing 133 unique enhancers. Top associations in the brain include ENSR00000320019 ($p = 1.31 \times 10^{-28}$ in amygdala), ENSR00000320042 ($p = 5.94 \times 10^{-26}$ in substantia nigra), ENSR00000195227 ($p = 2.48 \times 10^{-21}$ in cortex), ENSR00000032823 ($p = 3.67 \times 10^{-16}$ in cerebellum), and ENSR00000195227 ($p = 3.24 \times 10^{-15}$ in hippocampus).

Next, we ran TWAS using PrediXcan models of canonical gene expression previously trained in the same cell and tissue types[14]. Compared with the FUSION-based gene expression models, which included 10,669 unique genes across two brain regions (204 of which were associated with SCZ), the PrediXcan-based models included 26,133 unique genes expressed across 49 cell and tissue types, including 17,843 genes expressed in one or more brain regions. Across the cell and tissue types, we identified 2755 gene-tissue associations outside of the MHC region that reached genome-wide significance ($p < 1.45 \times 10^{-7}$), including 498 in the brain. Top hits within the brain include *PRSS16* ($p = 3.53 \times 10^{-29}$ in cerebellar hemisphere and $p = 5.30 \times 10^{-26}$ in cerebellum), *HFE* ($p = 3.71 \times 10^{-27}$ in cortex), *BTN3A3* ($p = 1.58 \times 10^{-26}$ in cortex), *CNNM2* ($p = 4.72 \times 10^{-24}$ in frontal cortex BA9), and *GNL3* ($p = 8.15 \times 10^{-24}$ in cortex), each of which has been previously implicated as a SCZ risk gene[31-36].

We investigated whether the original GWAS could have indirectly implicated these TWAS associations. Overall, of the 206 independent non-MHC loci that reached genome-wide significance in the original GWAS ($p < 5 \times 10^{-8}$), 6 had only an eRNA association, and 92 had only a

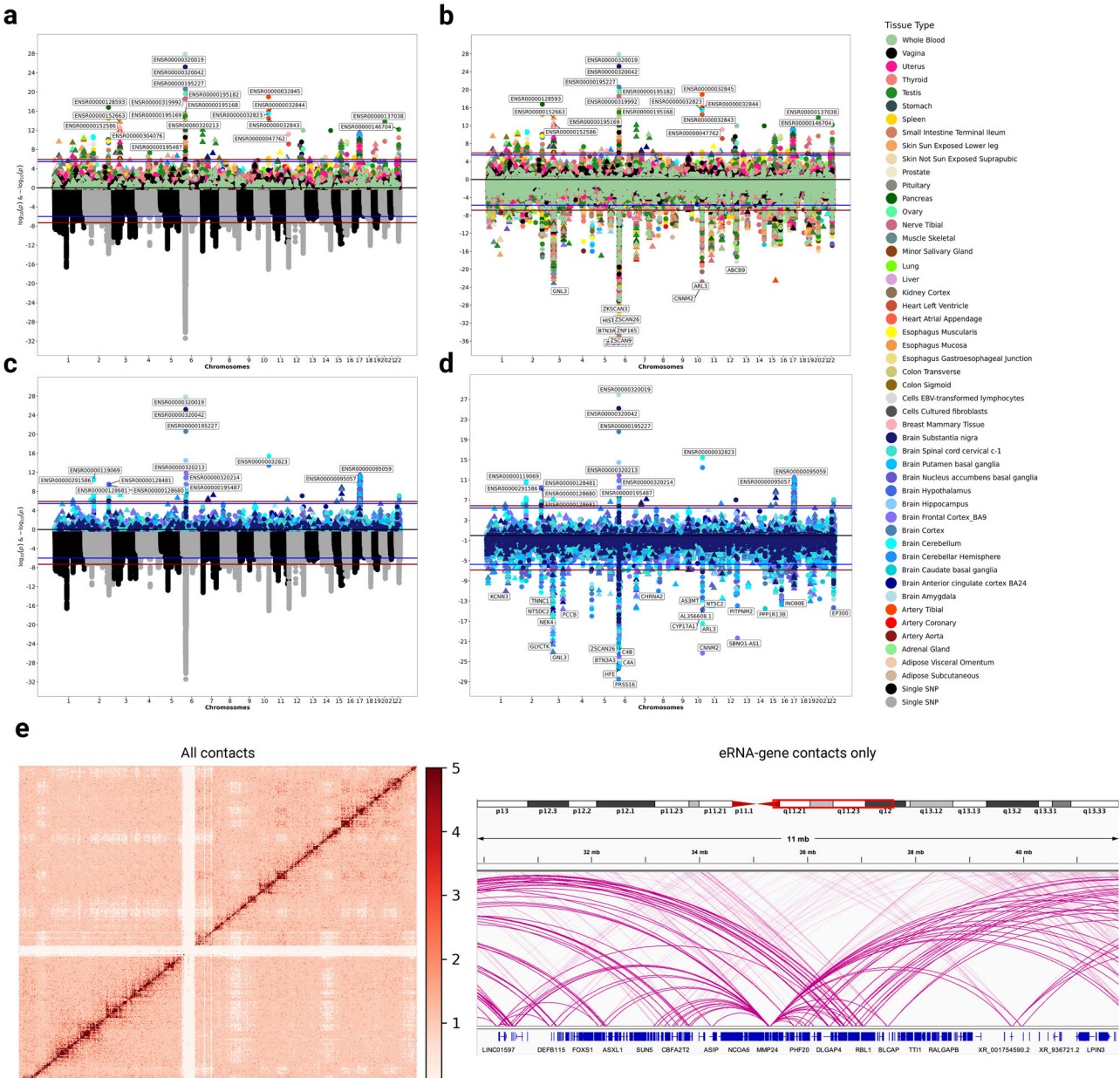

**Fig. 4 | eRNA-based TWAS of schizophrenia with corresponding GWAS and canonical gene TWAS associations.** TWAS was performed using the summary statistics from a logistic regression-based GWAS of schizophrenia[30] (plotted on the bottom half of **a** and **c** in grayscale). The red line in these plots indicates the Bonferroni-corrected genome-wide significance threshold (p = 5 x 10⁻⁸), and the blue line indicates suggestive significance (p = 1 x 10⁻⁶). The eRNA-based TWAS (plotted on the upper half of each plot) included 40,749 eRNA-tissue pairs (14,471 unique transcripts), the Bonferroni-corrected p-value threshold used for genome-wide significance was 1.23 x 10⁻⁶ (plotted in red) and 3.46 × 10⁻⁶ (plotted in blue). Because the canonical-gene-based TWAS (plotted on the lower half of **b** and **d**) included 344,814 gene-tissue pairs (26,138 unique transcripts), the Bonferroni-corrected p-value threshold used for genome-wide significance was 1.45 × 10⁻⁷

(plotted in red) and 1.91 × 10⁻⁶ (plotted in blue) for suggestive significance. **a** The full set of eRNA TWAS results for all 49 GTEx tissues plotted to mirror the GWAS results. **b** eRNA-based and canonical-gene-based TWAS results across all 49 tissues. **c** The eRNA-based TWAS results from 13 brain-derived tissues plotted against the GWAS results. **d** Brain-based eRNA and canonical-gene TWAS results. **e** Hi-C contact data (10 kb resolution) from primary astrocytes of the cerebellum was used to identify enhancer-gene contacts in the brain. Significant transcribed enhancer and canonical gene associations from TWAS in physical contact were tested for putative causality. The heatmap on the left depicts all Hi-C contacts (from chr20) prior to filtering, with the color scale corresponding to normalized contact frequency. The righthand plot shows a subset of those contacts between a transcribed enhancer and canonical gene.

canonical gene association. A further 45 loci had both an eRNA and canonical gene association in at least one tissue type, indicating a notably high degree of mirroring between eRNA and canonical-gene-based TWAS associations (Fig. 4b and d). Collectively, the total number of eRNA TWAS associations in the brain (114) could not be fully accounted for by the findings from the original GWAS, indicating the presence of independent eRNA associations with disease.

## Mendelian randomization illuminates causal mechanisms underlying SCZ risk

We utilized Mendelian randomization (MR) to investigate the causality of the eRNA and canonical gene associations with SCZ risk. Specifically, we conducted this analysis for each genome-wide significant association (p < 1.23 × 10⁻⁶ for eRNA associations and p < 1.45 × 10⁻⁷ for canonical gene associations).

Of the 392 significant TWAS eRNA associations outside of the MHC region, 222 (56.63%) showed evidence of causal effect on SCZ from MR, representing 104 unique transcribed enhancers (Supplementary Data 4). Of these putatively causal eRNA effects, 64 were in the brain, representing 34 unique enhancers. Out of the 2755 significant gene associations identified in the canonical-gene-based TWAS, 1297 (47.07%) were predicted to be causal, representing 396 unique genes (Supplementary Data 5). 226 of these putatively causal effects were in the brain, representing 130 unique genes.

The presence of putatively causal enhancers and canonical genes in a disease-associated locus raises the question of which transcript (eRNA or canonical gene) was causally "upstream" in the regulatory network. We asked whether an eRNA's causal effect on SCZ risk was due to the eRNA expression's influence on contact with a canonical gene in *cis* or, alternatively, if the eRNA association was not mediated through the expression of a canonical gene. To evaluate physical contact, we leveraged high-resolution enhancer-gene contact maps captured by Hi-C in astrocytes of the cerebellum. In brain, we identified 14 causal (based on MR) transcribed enhancers that were in physical contact with a causal canonical gene, representing 10 unique enhancer-gene pairs that were composed of four unique eRNAs and nine unique canonical genes (Supplementary Data 6). Additionally, we identified 57 additional causal enhancer-tissue pairs that were not in contact with a causal canonical gene in the Hi-C data, representing 33 unique eRNAs. Finally, we found 232 causal canonical gene-tissue pairs in the brain that were not in contact with a causal eRNA, representing 127 unique genes. The low proportion of physical interaction between causal enhancers and causal canonical genes (only ~4.6% of all causal transcript-tissue pairs) suggests that the mechanisms of eRNAs and genes in the context of SCZ risk are largely independent.

## Exploring the epigenomic context of causal eRNAs associated with SCZ

Several potential mechanisms of eRNA activity have previously been described in the literature[2], which can largely be bisected into two main classes: contact-dependent and contact-independent mechanisms. Under the contact-dependent model, eRNAs interact with the cohesin complex to promote interactions between the transcribed enhancer and nearby genes[5,37]. Under a contact-independent model, by contrast, transcribed eRNAs recruit transcription factors and other important chromatin modifiers to mediate chromatin state and activity level. For example, eRNA expression has been shown to increase recruitment of transcription factors such as YY1[6,38]. They can also interact with and recruit histone acetyltransferases CBP and p300 to increase H3K27ac[39,40]. Transcribed enhancers may also interact with the PRC2 complex, inhibiting deposition of the repressive mark H3K27me3[40,41]. The resulting open chromatin state can then facilitate assembly of super enhancer complexes and/or transcription of nearby genes[42].

To explore by which of these proposed mechanisms (contact-dependent or contact-independent) the brain-specific, causal eRNAs might influence SCZ risk, we utilized chromatin accessibility profiles (ATAC-seq and DNase-seq), as well as ChIP-seq of functionally informative histone modifications (H3K27ac, H3K27me3, and H3K4me1) and chromatin-associated proteins (RAD21, SMC3, CTCF, and EP300) in the SK-N-SH neuronal cell line to illuminate the regulatory landscape of these disease-associated transcripts. If these causal eRNAs influence SCZ risk via direct mediation of enhancer-gene interactions, we would expect to observe a strong enrichment of either cohesin complex subcomponents RAD21 and SMC3 or CTCF (Fig. 5a). In fact, however, of the 34 unique causal eRNAs tested, we observed RAD21 enrichment in only three (~9%), SMC enrichment in only one (~3%), and CTCF enrichment in only two (6%) (Supplementary Fig. 10a). Notably, we did not find enrichment of any of these chromatin contact-associated proteins within the subset of these eRNAs in 3D contact with a causal

gene. Based on this low enrichment of chromatin contact-associated proteins, in addition to the overall low proportion of causal eRNAs in physical contact with a causal canonical gene, we conclude that these results do not support the contact mediation model as a plausible explanation for eRNA associations with SCZ risk.

We next explored an alternative model to explain the potential mechanism by which the causal eRNAs mediate risk for SCZ. If causal eRNA expression plays a role in maintaining an open chromatin state, we should expect to observe an enrichment of ATAC-seq and DNase-seq peaks, as well as EP300 and H3K27ac, in addition to a depletion of H3K27me3. If there is super enhancer assembly in these regions, we would also expect to observe enrichment of histone mark H3K4me1. In concordance with these expectations, we observed overlap with either ATAC-seq or DNase-seq peaks in 12 (~35%) of the 34 of the causal eRNA regions, EP300 or H3K27ac enrichment in seven (~21%), and H3K4me1 enrichment in six (~18%) (Supplementary Fig. 10b). In total, we observed enrichment for at least one of these open chromatin-associated marks in 25 (~74%) of the 34 causal eRNAs. Notably, H3K27me3 was completely absent from these same causal enhancer regions. These results collectively support the contact-independent model of eRNA activity, in which causal transcribed enhancers influence SCZ risk via their role in establishing an open chromatin state.

## Motif enrichment analysis within SCZ-associated eRNAs
In addition to recruitment of chromatin modifiers, contact-independent eRNAs can also "capture" transcription factors (TFs), increasing TF occupancy at the transcribed enhancer[38]. To determine whether SCZ-associated enhancers were enriched for specific transcription factor binding sites, we performed motif enrichment analysis for all putative causal eRNA associations in brain (Supplementary Data 7 and 8). Across these 34 unique sequences, we observed motif enrichment (FDR < 0.05) for 135 TFs (Fig. 5b), several of which (EGR2[43–45], SOX10[46,47], TCF4/ITF2[48,49], and SP4[50,51]) have previously been implicated in SCZ. Enrichment of these factors' binding motifs at SCZ-associated eRNAs suggests a possible *trans* mechanism of eRNA-mediated SCZ risk, by which genetically regulated eRNA expression levels influence TF occupancy at disease-relevant enhancers.

## Phenome-wide TWAS of eRNAs in the UK Biobank
We performed TWAS across 4,671 complex traits using the eRNA models. We identified 467 traits with at least one genome-wide significant ($p < 2.60 \times 10^{-10}$) eRNA association (Fig. 6a), representing 88,348 significant eRNA-tissue associations (Supplementary Data 9-11).

Traits most highly enriched for significant eRNA associations include mean signal-to-noise ratio, a measure of hearing ability (1430 significant eRNA-tissue associations, lead association $p = 6.49 \times 10^{-143}$), as well as sitting height, a well-studied phenotype known to be highly polygenic[52–54] (1094 significant eRNA-tissue associations, lead association $p = 1.71 \times 10^{-106}$). Additional notable phenotypes highly enriched for significant eRNA-tissue associations included blood cell traits such as platelet distribution width (676 associations), neutrophil percentage (625 associations), hemoglobin concentration (583 associations), lymphocyte count (562 associations), and leukocyte count (394 associations); as well as smoking status (554 associations), cholesterol (536 associations), manifestations of mania or irritability (374 associations), weight (357 associations), and skin color (221 associations). We present the TWAS summary statistics as a comprehensive, tissue-wide resource of eRNA associations across the human phenome.

## Enhancer perturbation links eRNAs to canonical target gene expression
Epigenomic analysis of causal SCZ-associated eRNAs supported a context-independent, rather than contact-dependent model. These data alone, however, cannot fully explain the underlying mechanisms by which eRNA expression influences disease risk. To investigate

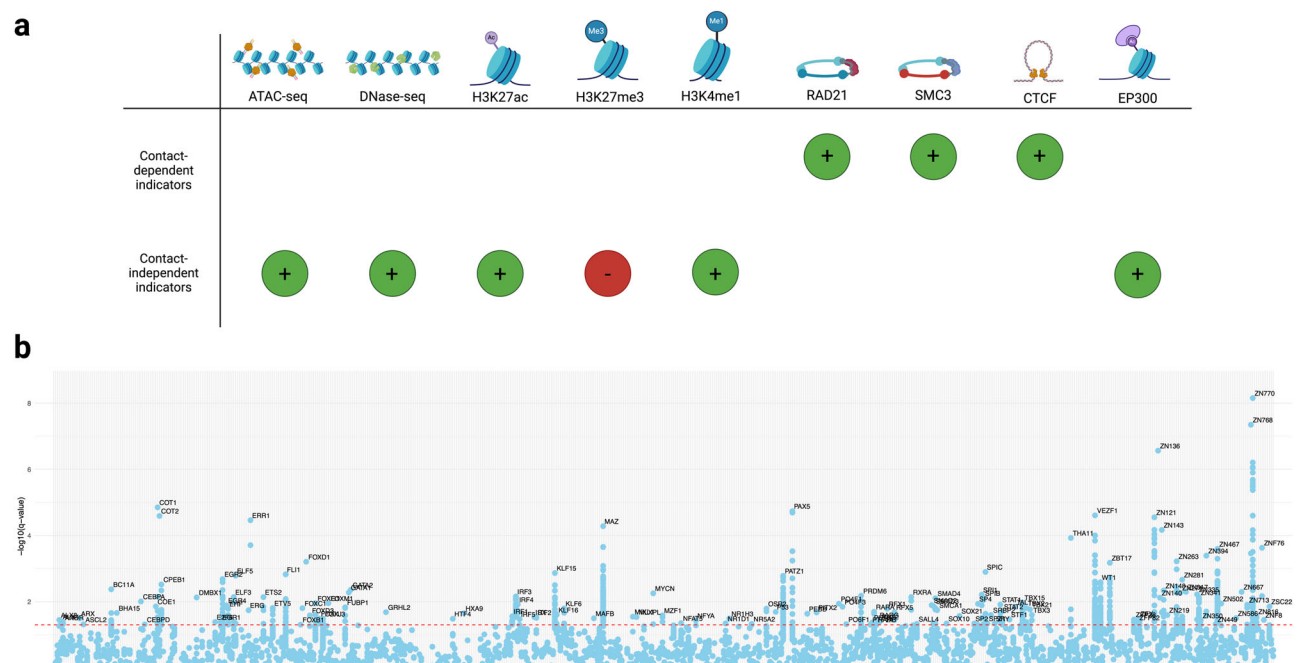

**Fig. 5 | Functional analysis of causal eRNAs suggests that these transcripts mediate SCZ risk via contact-independent rather than contact-dependent mechanisms. a** We hypothesized that if the causal eRNAs influenced SCZ risk via contact-dependent mechanisms, we should observe positive enrichment of cohesin complex sub-components RAD21 and SMC3 or CTCF. If the causal eRNAs influenced SCZ risk via contact-independent mechanisms, we hypothesized that one should observe enrichment of chromatin accessibility, histone marks H3K27ac and H3K4me1, and histone acetyltransferase EP300, as well as a depletion of H3K27me3. We would also expect an absence of RAD21, SMC3, and CTCF. Our functional analyses supported a contact-independent model. Expected enrichment is indicated with a + and marked in green, while expected depletion is indicated with a – and marked in red. **b** Transcription factor binding motif enrichment analysis using log likelihood ratio identified motifs for 135 unique factors that were significantly enriched in the sequences of the causal SCZ-associated eRNAs. Several of these TFs (EGR2[43–45], SOX10[46,47], TCF4/ITF2[48,49], and SP4[50,51]) have previously been implicated in SCZ. The significance threshold (FDR < 0.05) is indicated in red. Source data are provided as a Source Data file. a was created in BioRender[118].

whether disease-associated eRNAs have an effect on canonical gene expression, we utilized CRISPR perturbation assays in the K562 cell line targeting 109 complex trait-associated, transcribed enhancers identified by TWAS (Supplementary Data 12-13). Of these 109 enhancers, we identified 14 (12.84%) for which CRISPR perturbation resulted in a significant change in expression of a corresponding gene. Notably, we did not observe Hi-C contacts between any of these 14 eRNAs and their target gene(s). We also performed eQTL mapping in whole blood for both eRNAs and canonical genes. Among the 22 unique eRNA-gene pairs identified using CRISPR perturbation, we observed only one mapped SNP eQTL (FDR < 0.1) that was shared by both the eRNA and canonical gene in a pair. Thus, these eRNAs appear to mediate canonical gene expression independently of chromatin contacts or shared SNP eQTLs.

Among some of the disease-relevant eRNAs linked to canonical gene expression were ENSR00000013481 and ENSR00000032851, both associated with SCZ; ENSR00000117322, associated with manifestations of mania or irritability; and ENSR00000320642, associated with hypertension. Perturbation of SCZ-associated eRNAs ENSR00000013481 and ENSR00000032851 resulted in decreased expression of *VPS45* (fold change = 0.75, $p = 1.21 \times 10^{-3}$) and *NT5C2* (fold change = 0.74, $p = 7.66 \times 10^{-4}$), respectively. Both genes have previously been implicated in SCZ and reach significance in our canonical gene-based TWAS of SCZ[55–57].

In addition to the two SCZ-associated eRNAs, perturbation of ENSR00000117322 (associated with Manifestations of mania or irritability) resulted in decreased expression of the SCZ-associated gene *RTN4* (fold change = 0.83, $p = 6.91 \times 10^{-9}$), while perturbation of hypertension-associated eRNA ENSR00000320642 resulted in decreased *MRPS10* expression (fold change = 0.82, $p = 1.42 \times 10^{-8}$). This gene codes for a mitochondrial ribosomal protein and has previously been linked to cardiac disorders[58–60]. Neither of these eRNA-linked genes reached significance in a canonical gene-based TWAS.

## Inclusion of eRNAs increases GWAS signals explained by an eQTL

Recent work reports systematic differences in identified genetic effects on complex traits (through GWAS) and gene expression (via eQTL mapping)[61], so that most GWAS signals are not explained by known eQTLs[62–64]. A number of strategies may bridge this colocalization gap, including utilizing larger sample sizes for increased eQTL mapping power, mapping eQTLs in more expansive sets of cell and tissue types, as well as considering QTLs modulating chromatin structure and splicing[61]. We investigated whether the use of eRNA eQTLs might improve GWAS signal interpretability.

Across all tissues, we observed a mean Jaccard similarity index of 0.05 between eRNA eQTLs (Methods) and canonical gene eQTLs, indicating a notably low overlap between the eQTL two sub-classes (Supplementary Fig. 11). Overlap was highest in the testis (Jaccard similarity index = 0.09) and lowest in brain putamen (Jaccard similarity index = 0.03).

The previously reported systematic discordance between GWAS signals and canonical gene eQTLs relies partly on the observation that eQTLs tend to be proximal to genes, while GWAS signals are more distal[61]. We found that, like GWAS hits, eQTLs specific to eRNAs were significantly more distal to a TSS than canonical gene eQTLs (median distance 41.93 kb versus 14.36 kb, Mann-Whitney U test $p < 2.2 \times 10^{-16}$).

We finally aimed to explore whether considering eRNA eQTLs might help to bridge the gap between eQTLs and GWAS signals. Using the same sets of mapped eQTLs, we performed colocalization analysis[65]

**a**

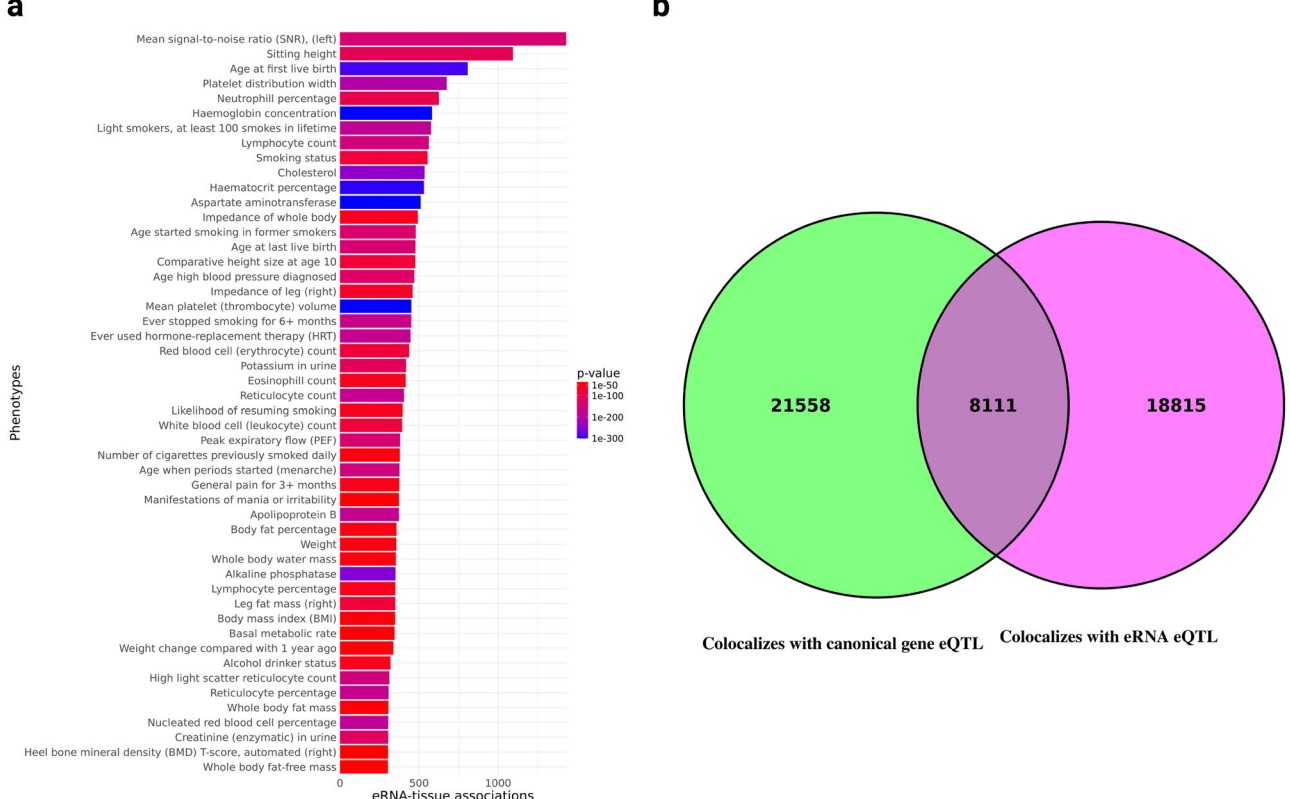

**b**

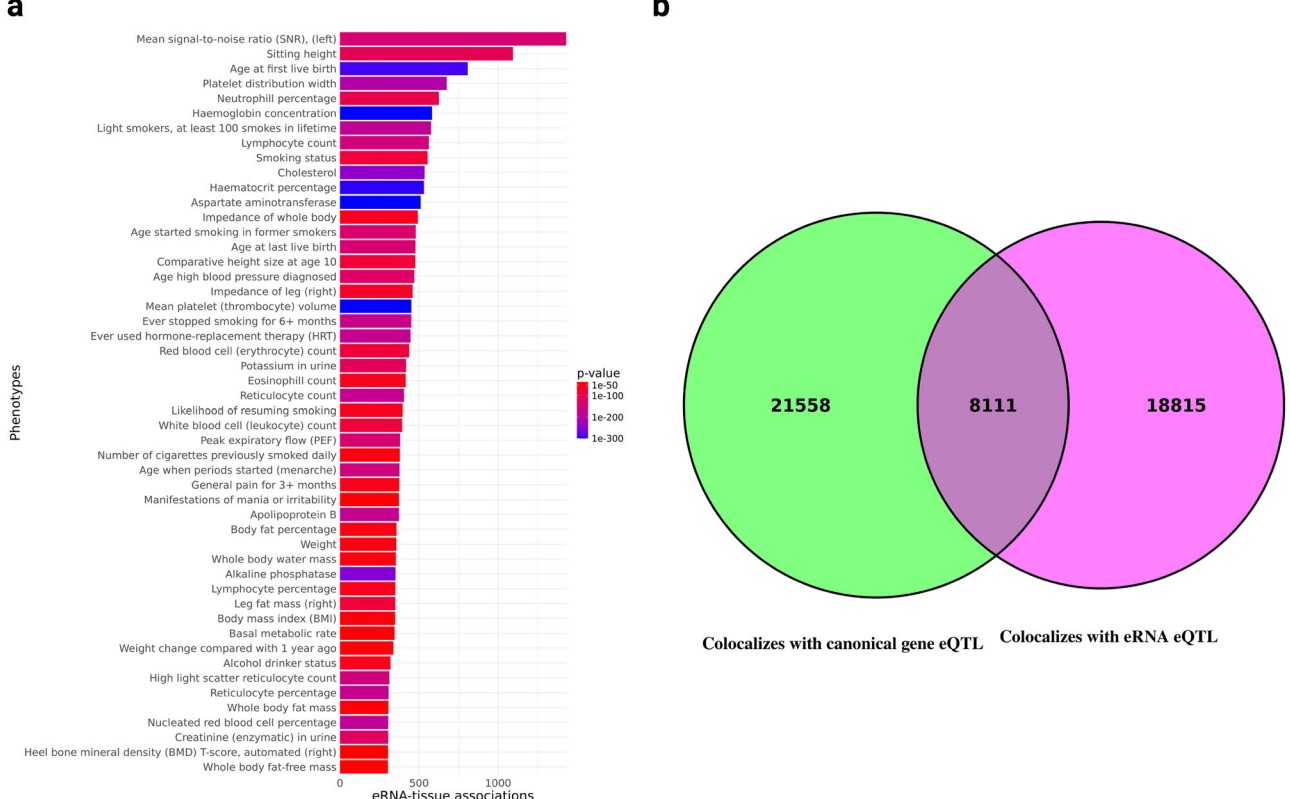

**Fig. 6 | eRNA eQTLs are associated with complex traits across the phenome and help to explain 63% more GWAS signals than canonical gene eQTLs alone.**
**a** Depicted are the top 50 heritable UK Biobank traits ranked by number of significant (p < 2.60 x 10⁻¹⁰) eRNA-tissue TWAS associations. The color for each phenotype corresponds to the p-value of its most significant eRNA association. **b** Using

a Bayesian framework, we identified 18,815 genome-wide significant (p < 5 × 10⁻⁸) GWAS signals within the UK Biobank that only colocalized (posterior probability > 0.7) with an eRNA eQTL, representing a 63% increase over the number of independent associations that can be explained by canonical gene eQTLs alone. Source data are provided as a Source Data file.

across 996,391 independent, genome-wide significant ($p < 5 \times 10^{-8}$) associations from the UK Biobank[10], representing 4,671 complex traits. We identified 26,926 GWAS associations that colocalized with an eRNA eQTL (posterior probability ≥ 0.7) in at least one tissue, and a further 29,669 GWAS associations colocalized with a canonical gene eQTL (Supplementary Data 14-15). In total, there were 48,484 GWAS associations that colocalized with either an eRNA or canonical gene eQTL, with 8,111 (16.75%) colocalizing with both and the remaining 40,303 (83.25%) colocalizing with only one of the eQTL sub-classes. Notably, we identified 18,815 GWAS signals exclusively colocalizing with an eRNA eQTL, resulting in a substantial 63% increase in the total number of GWAS signals with shared causal variants with eRNAs (Fig. 6b). Collectively, these results show that eRNA analysis can substantially improve our ability to mechanistically interpret GWAS associations.

## Discussion

In this work, we develop in silico models of genetically regulated eRNA expression and present several notable findings, from chromatin contact frequency prediction to eRNA-based TWAS. Across all tissues, eRNA expression showed a significantly higher mean number of SNP eQTLs per transcript versus canonical gene expression. We propose two possible explanations. Since enhancers have a high degree of tissue-specific and context dependent activity[66–70], eRNA expression likewise may be under a higher degree of fine-tuned genetic control than protein-coding genes. Alternatively, the high number of SNP eQTLs per eRNA could also be due to the rapid evolution of enhancer sequences relative to those of canonical genes[71]. Enhancers, as non-coding elements in the genome, may exhibit a higher tolerance to mutation than a typical protein-coding gene.

We imputed genetically determined eRNA expression and canonical gene expression across more than 70,000 individuals in a large-scale biobank, BioVU. The mean GReX values generated for two of the tissues (whole blood and cerebellum) were subsequently used to train two independent neural networks to predict 3D chromatin contact frequency. Previous work has demonstrated that genomic distance, Hi-C, and other 1D epigenomic datasets can be used to predict chromatin contact[27,72–77]. Here, we demonstrate that GReX, too, is predictive of chromatin contact frequency.

Our deep learning models, trained in cerebellum and whole blood, predicted tissue-dependent contact frequency with substantially greater performance than both the corresponding linear and non-linear regression models, which showed limited predictive power. Notably, the whole blood-based GReX model retained comparable predictive accuracy in a genome-wide nuclear run-on dataset from K562, indicating that the model captures key patterns underlying the relationship between expression and contact frequency.

While the cerebellum-based (deep learning) model appeared to be portable to a tissue not observed during training ($R^2 = 0.18$ in whole blood), the whole blood-based model had limited predictive power ($R^2 = 0.01$ in cerebellum). Two possible explanations may explain this discrepancy in cross-tissue performance. First, relative to other tissue types, whole blood is highly heterogeneous, composed of a variety of different cell types and metabolites whose relative proportions can be highly variable within a given sample[78–80]. Second, gene expression in whole blood is highly dynamic relative to other tissues, with rapid changes in expression patterns induced in response to a wide range of environmental stimuli, including even the changing of the seasons[81]. The lower stochasticity of cerebellum expression could account, at

least in part, for the greater cross-tissue generalizability of the cerebellum-based model, capturing fundamental patterns in the relationship between GReX and contact frequency.

The use of functional information enhances our ability to interpret the phenotypic consequences of complex genetic variation[82]. Because of cost, time, and limited sample volumes, it is often not feasible to generate rich omics datasets, at a biobank scale. In recent years, approaches such as PrediXcan[13] have offered a means of circumventing this barrier in silico, allowing researchers to impute genetically regulated expression, both at the gene level and even for individual splice isoforms[83,84], using only germline variation. Generating a high-resolution tissue-specific chromatin contact map remains methodologically challenging due to the quadratic scale of the data. Here, we have utilized the GReX models to gain further insights into 3D chromatin organization, coupling genetically determined transcription and contact maps.

Because the vast majority of genetic associations with complex disease are within non-coding regions of the genome[85], TWAS have become a valuable tool for inferring the relevant gene(s) in a GWAS locus[86]. A previous TWAS of schizophrenia, for example, found that the disease-associated, non-coding genetic variation identified at locus 16p11.2 results in increased expression of the gene *MAPK3*[87]. The mechanistic contribution of eRNAs to disease risk remains to be elucidated. Recent work focused on neuropsychiatric phenotypes strongly suggested that enhancer expression quantitative trait loci (EeQTLs), or SNPs regulating the transcription of enhancers in the brain, can be used to investigate disease mechanisms[8]. The authors concluded that most of the disease heritability captured by eRNAs is independent of that explained by eQTLs of protein-coding genes. Based on these initial findings in the brain, eRNA expression in other tissues is likely to be a broadly important contributor to disease biology across the human phenome. We therefore make available eRNA-based models trained in a large collection of tissues to facilitate downstream genomic applications.

Using eRNA-based TWAS, we identified 392 eRNAs associated with schizophrenia, representing 104 unique eRNAs. This same approach was then applied to a set of phenome-wide GWAS traits in the UK Biobank[10], which we present as a significant curated resource for elucidating the phenomic consequences of enhancers. Among a sample of 109 complex trait-associated eRNAs, experimental perturbation resulted in a change in canonical gene expression for ~12%. Because these CRISPR perturbations results are derived from the leukemic K562 cell line, it is possible that a larger proportion of these eRNAs may regulate canonical gene expression in other cell types. Notably, of those that were linked to a canonical gene in K562, however, no eRNA-gene pairs showed evidence of physical interaction in Hi-C contact data. We also did not observe eQTL sharing between eRNAs and canonical genes across linked pairs. These results suggest that rather than a pleiotropic model, in which the same set of eQTLs regulate GReX of both eRNA and canonical gene transcription at a given locus, eRNA and canonical GReX are under largely independent mechanisms of genetic control.

Due to the low frequency of observed contacts between complex trait-associated eRNAs and canonical genes, we explored the epigenomic context of these enhancer sequences to further illuminate potential mechanisms by which these causal eRNAs mediate disease risk. Among causal eRNAs associated with SCZ, we observed high enrichment for functional features associated with open chromatin state and relatively low enrichment of features associated with chromatin loop formation. One key caveat of these functional analyses is that the ChIP-seq assays utilized, particularly those of EP300, RAD21, SMC3, and CTCF, profile protein binding to DNA rather than RNA. Due to the unavailability of RIP-seq and CLIP-seq profiles of these factors in neuronal cells, we are unable to directly assess eRNA-protein interactions and must instead use DNA binding as a proxy. A more comprehensive characterization of the epigenomic consequences of eRNA perturbation represents an exciting direction for future work.

The functional results that we do present, however, along with significant binding motif enrichment within causal eRNAs for SCZ-associated TFs, strongly suggest that these causal disease-associated eRNA act via contact-dependent rather than contact-independent mechanisms. Under this model, genetically regulated eRNA expression levels influence the recruitment of both chromatin modifiers and TF occupancy at corresponding enhancers. Genetic control of these contact-independent regulatory mechanisms would likely influence canonical gene expression downstream, as supported by CRISPR perturbation of some of these disease-associated enhancers.

Finally, within the comprehensive catalog of disease-associated eRNAs in the UK Biobank, we show that the use of eRNAs holds promise in closing the so-called colocalization gap with GWAS traits, facilitating a 63% increase in the number of significant GWAS associations (from the UK Biobank) that can be explained by eQTLs relative to the use of canonical genes alone.

Our study has some important limitations. The underlying eRNA expression data on which our GReX models are trained are enriched for polyadenylated transcripts[88]. While 1D RNAs are similar in structure to mRNAs and lncRNAs (long, spliced, poly-adenylated, and uni-directionally transcribed), 2D eRNAs, which constitute the majority of eRNAs, are structurally distinct from canonical gene RNAs (short, unspliced, non-polyadenylated, and bidirectionally transcribed). Thus, our eRNA dataset likely underrepresents the full set of 2D eRNAs detectable using a direct quantification approach such as a nuclear run-on assay. However, transcript length characterization of our dataset suggests that over 90% of eRNA transcripts included in these models exhibit characteristics of 2D eRNAs, which comprise the majority of naturally occurring eRNAs[15]. In addition, approximately 85% of GTEx samples were derived from individuals of European ancestry[89]. Thus, the generalizability of these models to non-European ancestries remains to be investigated[90]. Furthermore, the neural network approach we implemented for contact frequency prediction captures only a proportion of the variation in 3D chromatin organization. As multiple enhancers may jointly regulate a target gene, additional local features in the form of GReX may further enhance the contact map prediction performance. Nevertheless, as demonstrated here, these eRNA models provide a powerful set of tools to explore a relatively understudied component of the genome that is ripe for new discoveries.

## Methods

### Ethics

This research includes genetic data from deceased human individuals from the GTEx Project[9]. The protected data for the GTEx Project (for example, genotype and RNA-seq data) are available via access request to dbGaP accession no. phs000424.v8.p2.

Analyses using BioVU data comply with all ethical regulations as approved by the Vanderbilt University Medical Center institutional review board (IRB 151187 and IRB 160372). All requests for raw (for example, genotype and phenotype) data and materials are reviewed by Vanderbilt University Medical Center to determine whether the request is subject to any intellectual property or confidentiality obligations. For example, patient-related data not included in the paper may be subject to patient confidentiality. Any such data and materials that can be shared will be released via a material transfer agreement.

### In silico genetic models of eRNA expression

The Human enhancer RNA Atlas (HeRA)[91] (https://hanlab.uth.edu/HeRA/) provides a publicly available resource of eRNA expression values processed from the raw GTEx[9,92] RNA-seq reads. These expression values were first normalized using the first 5 principal components, the first 5 PEER covariates[93], age, sex, and sequencing platform. Next, using an implementation of PrediXcan[13] provided by MR-JTI[14], we trained models for each of the 49 GTEx cell and tissue types using

whole genome sequencing (WGS) and the corresponding quantified eRNA expression data generated from 507 donors. PrediXcan models of canonical gene expression were previously trained using the implementation provided in the MR-JTI repository and retrieved from Zenodo under accession code 3842289 (JTI)[94].

## Training set for nuclear run-on-based chromatin contact prediction

Because both eRNAs and canonical genes are 5'-capped[95], K562 GRO-cap dataset enriched for capped RNAs was obtained from the ENCODE Data Portal[96] (accession number ENCSR363AKK) in bigWig format. Files quantifying transcription from the plus and minus strands were combined and converted to bedGraph format. Using bedtools[97] (v2.30.0), transcripts in the nuclear run-on assay that overlapped with a known human eRNAs annotated by the ENSEMBL[98], FANTOM5[99], or Roadmap[96] consortia or a known GENCODE[100,101] (v32) gene were identified. Transcript counts annotated with the same eRNA or gene were combined. Prior to model training, these final expression values were log1p-normalized.

## Training set for GReX-based chromatin contact prediction

The trained models of eRNA and canonical gene expression were utilized to impute GReX in 72,828 BioVU samples across all 49 cell and tissue types. The BioVU data consisted of individuals of European ancestry (31,861 males, 40,584 females, and 383 with unknown sex) genotyped on the Illumina MEGA array, followed by genotype imputation using the HRC panel. Age among individuals in the sample ranged from 0-90, with a median age of 56.

High-resolution Hi-C datasets that had been generated in the K562 leukemic cell line (accession number 4DNFI18UHVRO) and astrocytes of the cerebellum (accession number 4DNFIWCAQUIK) were retrieved from the 4D Nucleome[19] Data Portal (https://data.4dnucleome.org) in mcool format. Raw sequencing reads underwent initial processing, contact matrix aggregation, and normalization using the gold standard Hi-C processing pipeline detailed at https://data.4dnucleome.org/resources/data-analysis/hi_c-processing-pipeline. The Hi-C dataset representing whole blood (K562) included 907,136,828 filtered reads, while the cerebellum-based dataset included 428,475,763. Both datasets showed similar quality control metrics, such as cis/trans ratio, % long-range intrachromosomal reads, and very good convergence (Supplementary Data 16). Contacts were normalized using the ICE (iterative correction and eigenvalue decomposition) algorithm[102].

With the processed mcool file, Cooler (v0.8.2)[103] was utilized to export contacts at a 10 kb resolution using the *cooler dump* function, and an R script was written to convert these outputs to BEDPE format (see GitHub[104]). We defined an initial set of contact regions as those with one or more normalized contact in each Hi-C dataset. We then filtered these contact pairs down to a subset in which the respective 10 kb contact regions overlapped with either two annotated eRNAs or an eRNA and canonical gene. The eRNA annotations used were from the same collection utilized by the authors of the Human enhancer RNA Atlas (HeRA)[91], and consisted of human eRNAs annotated by the ENSEMBL[98], FANTOM5[99], and Roadmap[96] consortia. Canonical gene annotations were obtained from GENCODE (v32)[100,101].

These contact data, along with mean eRNA and canonical gene GReX values for whole blood and cerebellum in BioVU, respectively, comprised the training set. The resulting dataset contained 85,630 unique genome-wide enhancer-gene contact pairs in whole blood and 95,701 in cerebellum. Because the genomic distance distribution was left-skewed, these values underwent log1p transformation.

## Linear regression models of chromatin contact frequency

Using scikit-learn[105] (v1.2.2), two linear regression models were fit to predict contact frequency for enhancer-enhancer and enhancer-gene pairs, one based on genomic distance between the two elements and the other based on GReX of the respective eRNA and gene. An 80/20 train-test split was used, meaning that the whole blood training set consisted of 68,504 unique contact pairs, while the held-out test set was composed of 17,126. The number of contact pairs in the cerebellum training and test sets were 76,560 and 19,141, respectively.

## Non-linear models of chromatin contact frequency

Using scikit-learn[105] (v1.2.2), four non-linear models (polynomial regression, random forest regression, support vector regression, and gradient boosting regression) were fit to predict enhancer-enhancer and enhancer-gene contact frequency using GReX data from both whole blood and cerebellum. An 80/20 train-validation split was used with grid search to select the optimal set of hyperparameters for each model.

Polynomial regression grid search utilized degrees ranging from 1 to 10. For random forest regression grid search, we tested tree numbers ranging from 10 to 100. Support vector regression grid search tested epsilon values ranging from 0.1 to 1. Gradient boosting regression grid search tested boosting stage numbers ranging from 10 to 300. $R^2$ was used as the selection metric for determining each best-performing model.

## Training deep learning models based on eRNA and canonical gene GReX

All deep learning models were trained on an Nvidia Tesla K80 GPU running CUDA[106]. Using PyTorch[107] (v1.13.1), two fully connected neural networks were trained to predict enhancer-gene contact frequency. The first model was trained using solely the mean genetically regulated eRNA and canonical gene expression imputed in BioVU, while the second model also included the genomic distance between the two transcripts.

Both models were trained using five-fold cross validation, and hyperparameters were tuned using grid search. An 80/20 train-test split was used. During grid search, each network was iteratively optimized for the number of hidden neurons, number of hidden layers, batch size and number of training epochs, optimizer and learning rate, weight initialization in the hidden and output layers, hidden layer activation function, dropout, and weight constraint, and L1 and L2 regularization parameters. Models were trained using both $R^2$ and root mean squared error (RMSE) as the selection criterion and showed comparable results (Supplementary Fig. 12). Optimal models were ultimately selected based on maximization of prediction $R^2$.

Because this was a regression problem, mean squared error (MSE) served as the loss function:

$$\text{MSE} = \frac{1}{n}\sum_{i=1}^{n}\left(Y_i - \hat{Y}_i\right)^2 \qquad (1)$$

Here, $n$ is equal to the number of data points, $Y_i$ represents the observed outcome, and $\hat{Y}_i$ represents the predicted outcome. We assumed a combination of L1 and L2 regularization. Predictive performance was evaluated using the $R^2$ metric:

$$R^2 = 1 - \frac{\sum_{i=1}^{n}\left(\hat{Y}_i - Y_i\right)^2}{\sum_{i=1}^{n}\left(Y_i - \bar{Y}\right)^2} \qquad (2)$$

Here, $n$ is equal to the number of data points, $Y_i$ represents the observed outcome, $\hat{Y}_i$ represents the predicted outcome, and $\bar{y}_i$ represents the mean of the observed outcomes.

## SCZ PrediXcan TWAS

Using the eRNA models and previously published canonical gene expression models[14], S-PrediXcan[108], implemented in the MetaXcan GitHub repository (https://github.com/hakyimlab/MetaXcan), was utilized to perform summary statistics based TWAS on a recent GWAS

meta-analysis of schizophrenia[30]. Multiple testing was accounted for using a Bonferroni correction. Thus, for the eRNA-based TWAS, the p-value threshold for genome-wide significance was $1.23 \times 10^{-6}$ $\left(\frac{0.05}{40,749\,eRNA-tissue\,pairs}\right)$, while the threshold for suggestive significance was $3.46 \times 10^{-6}$ $\left(\frac{0.05}{14,471\,unique\,eRNAs}\right)$. Meanwhile, for the canonical gene-based TWAS, genome-wide significant and suggestive p-value thresholds of $1.45 \times 10^{-7}$ $\left(\frac{0.05}{344,814\,gene-tissue\,pairs}\right)$ and $1.91 \times 10^{-6}$ $\left(\frac{0.05}{26,138\,unique\,genes}\right)$, respectively, were used.

### Mendelian randomization of significant SCZ TWAS loci
Genome-wide significant associations from the respective eRNA-based and canonical gene-based TWAS were tested for causality using Mendelian randomization implemented in MR-JTI[14] (https://github.com/gamazonlab/MR-JTI). After testing each eRNA-tissue and canonical gene-tissue association for causality, the identified causal associations were localized to their respective genetic loci to assess which loci had a causal eRNA only, canonical gene only, or both for SCZ.

### Investigating potential functional mechanisms of causal eRNAs
ChIP-seq datasets generated in the SK-N-SH neuronal cell line were downloaded from the ENCODE Project[96,109] data portal (https://www.encodeproject.org) in narrowPeak format for H3K27ac (ENCFF362OBM), H3K27me3 (ENCFF277NRX), H3K4me1 (ENCFF580GTZ), RAD21 (ENCFF051ZRW), SMC3 (ENCFF756DQA), CTCF (ENCFF244QKO), and EP300 (ENCFF654KAP). ATAC-seq (ENCFF716JUM) and DNase-seq (ENCFF752OZB) datasets were also retrieved for the same cell line. All datasets based on hg38 were lifted over to hg19 using liftOver[110].

### Transcription factor binding motif enrichment
FIMO[111] (v5.4.1, https://meme-suite.org/meme/doc/fimo.html) was used to search for enrichment of HOCOMOCO[112] v11 human TF binding motifs in the set of 34 causal SCZ-associated eRNAs in brain. Default parameters were used, and motifs with a q-value (FDR) < 0.05 were considered to have significant enrichment.

### Phenome-wide eRNA-based TWAS in the UK Biobank
Using S-PrediXcan[108], the eRNA-based models were applied to GWAS summary statistics across 4,671 traits in the UK Biobank[10] to generate a comprehensive resource of tissue-specific enhancer associations. A genome-wide significant threshold of $p < 2.60 \times 10^{-10}$ was used, with the multiple testing threshold for genome-wide significance determined using Bonferroni correction:

$$p_{eRNA} = \frac{0.05/41,086\,eRNA-tissue\,pairs}{4671\,complex\,traits} \quad (3)$$

### Enhancer perturbation analysis for disease-associated eRNAs
A previously published enhancer perturbation dataset was obtained for the K562 cell line. Using CRISPRi, 5920 human enhancers were perturbed, and resulting changes in gene expression were measured[113]. Perturbed enhancer coordinates were converted to UCSC BED format, and enhancers that overlapped with a known eRNA annotated by the ENSEMBL[98], FANTOM5[99], or Roadmap[96] consortia were identified using bedtools[97] (v2.30.0, https://bedtools.readthedocs.io/en/latest/).

### eRNA and canonical gene eQTL mapping
The same genotype and normalized expression datasets used for eRNA and canonical gene-based GReX model training was used to perform eQTL mapping. *Cis*-eQTLs were mapped using Matrix eQTL[114] (v2.3).

### Comparing eRNA eQTLs and canonical gene eQTLs
For each tissue, the overlap between the eRNA eQTLs and the canonical gene eQTLs s was assessed using the Jaccard similarity index.

To assess the distance from a canonical gene, we used the human TSS coordinates from refTSS[115] version 4.1 (https://reftss.riken.jp/datafiles/4.1/human/refTSS_v4.1_human_coordinate.hg38.bed.txt.gz). The coordinates were lifted from hg38 to hg19 using liftOver[110] (http://hgdownload.soe.ucsc.edu/admin/exe/liftOver.gz). Distance from the closest TSS for all eRNA and canonical gene eQTLs was computed using bedtools (v2.30.0, https://bedtools.readthedocs.io/en/latest/).

### Colocalization of GWAS signals with eRNA and canonical gene eQTLs
Across 4671 complex traits in the UK Biobank[10], 996,391 independent, genome-wide significant ($p < 5 \times 10^{-8}$) associations were identified. For each significant association, all GWAS SNPs and SNP eQTLs at the corresponding locus were utilized to perform colocalization analysis using coloc[65] (v5.2.3). Colocalization was run in each of the 49 GTEx tissues for both eRNA eQTLs and canonical gene eQTLs. A GWAS signal was considered to colocalize with an eQTL if the posterior probability was >= 0.7 in at least one tissue.

### Reporting summary
Further information on research design is available in the Nature Portfolio Reporting Summary linked to this article.

## Data availability
The eRNA GReX models, along with TWAS, Mendelian randomization, and colocalization results generated in this study are have been deposited in Zenodo under the accession code 14027849[116]. The trained contact frequency models are deposited on GitHub (https://github.com/mjbetti/erna-grex) and Zenodo under accession code 14557414[104]. The previously published canonical gene GReX models are deposited in Zenodo under the accession code 3842289[94]. Nuclear run-on data from the K562 cell line are available from the ENCODE Portal[96] under accession code ENCSR363AKK. Hi-C datasets for K562 and astrocytes of the cerebellum are available from the 4D Nucleome Data Portal[19] under accession codes 4DNFI18UHVRO and 4DNFIWCAQUIK, respectively. ChIP-seq and chromatin accessibility datasets for SK-N-SH are available from the ENCODE Portal[96] under accession codes ENCFF362OBM, ENCFF277NRX, ENCFF580GTZ, ENCFF051ZRW, ENCFF756DQA, ENCFF244QKO, ENCFF654KAP, ENCFF716JUM, and ENCFF752OZB. The curated dataset of human TSS coordinates is available from refTSS[115] (https://reftss.riken.jp/datafiles/4.1/human/refTSS_v4.1_human_coordinate.hg38.bed.txt.gz). The quantified eRNA expression data used to train GReX models are available from HeRA[91] (https://hanlaboratory.com/HeRA/). Due to the nature of the GTEx donor consent agreement, raw genotype data from GTEx V8[9] individuals are available under restricted access. These data are available via access request to dbGaP accession no. phs000424.v10.p2 [https://www.ncbi.nlm.nih.gov/projects/gap/cgi-bin/study.cgi?study_id=phs000424.v10.p2]. Access to de-identified BioVU genotype data[18] requires a collaboration with a Vanderbilt faculty member. Investigators may contact BioVU support (biovu@vumc.org) for additional information or assistance in establishing a collaboration. All requests for BioVU data and materials are reviewed by Vanderbilt University Medical Center to determine whether the request is subject to any intellectual property or confidentiality obligations. Any such data and materials that can be shared will be released via a material transfer agreement. The compiled datasets used to train contact frequency models, TF binding motif enrichment results, perturbed TWAS eRNAs, and Hi-C QC metrics, and underlying data used to plot figures are available in the Supplementary Information/Source Data file. Source data are provided with this paper.

## Code availability

All code generated for this work is publicly accessible on GitHub (https://github.com/mjbetti/erna-grex). A permanent version of all code is deposited on Zenodo under accession code 14557414[104].

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

## Acknowledgements

This research was supported by National Institutes of Health (NIH) grants NHGRI R35HG010718 (E.R.G.), NHGRI R01HG011138 (E.R.G.), NIA AG068026 (E.R.G.), NIGMS R01GM140287 (E.R.G.), NIMH R01MH126459 (E.R.G.), NIA R56AG089926 (E.R.G.), and NIH/NCI U01CA253560 (M.C.A.). This research was supported by a gift to E.R.G. from the Scott Hamilton CARES Foundation (in honor of Carlee Vaughn).

## Author contributions

M.J.B. and E.R.G. designed the study. M.B. performed the data analysis, with contributions from P.L., and wrote the manuscript, with contributions from E.R.G. and M.C.A. E.R.G. supervised the study.

## Competing interests

Dr. Gamazon has performed consulting for Thryv Therapeutics. Dr. Gamazon is a co-inventor on patents for molecular signatures of cardiovascular phenotypes and metabolic health, and the use of RNAs as therapeutics and diagnostic biomarkers. This had no influence on the research presented in this study. The remaining authors declare no competing interests.
