## [Transparent Peer Review file · Nature Communications]

Genetically regulated eRNA expression predicts chromatin contact frequency and reveals genetic mechanisms at GWAS loci

Corresponding Author: Dr Eric Gamazon

Version 0:

Reviewer comments:

Reviewer #1

(Remarks to the Author)

In this work the authors dig into the eRNAs and show their importance in explaining the GWAS variants on diseases. They show that expression of eRNAs has reasonable predictive power in contact frequency prediction. They further show evidence from functional assay that reveals the mechanisms by which eRNAs work and find out that they are more likely to follow the trans model by means of recruiting the chromatin modifiers and TFs rather than the cis model of acting through loop formation between enhancers and genes. I think this work has done some good analyses of showing the importance of eRNAs and their probable mechanisms. However, there are some important points missing, which I have summarized in the following comments:

- How do you determine the number of enhancer-enhancer and enhancer-promoter interaction using Hi-C data? There are many ways to do so, such as using a statistical method like HiCDCPlus, which also considers distance as a feature, and thresholding based on false discovery rate to get only the statistically significant interactions. It is not clear how the authors come up with the E-E and E-P interactions.
- The author mention that in the case of cross-tissue generalization, the models trained on the whole blood cannot generalize to cerebellum data, but the models trained on the cerebellum data can generalize relatively well to the whole blood data. This also brings up a question about normalization of the data, as if the normalization is not done in a rigorous way, this issue can be seen. For example, the authors must make sure if their input data (eRNA expression) and output (E-E and E-P) have been normalized in a way that considers the library size and sequencing depth of all these assays. Hi-C data in particular could end up with inconsistent numbers of called loops (E-E or E-P) if they have very different sequencing depth.
- The authors show low negative correlation between the predicted contact frequency and genomic distance in Fig S4. I am also interested to see the same plot for the true contact frequency vs genomic distance.
- The authors mention "the mechanisms of eRNAs and genes in the context of SCZ risk are largely independent." Do you have any insights on how the causal enhancers play a role in disease? It is great that the authors explore some mechanisms by which these causal eRNAs work, and they find that the trans model is dominant, meaning that eRNAs help making the chromatin more accessible rather than playing a role on loop formation. However, it is less clear how such enhancers lead to a disease. For example, in the context of a gene regulatory network, these enhancers might affect a cis gene/TF that might consequently affect their target genes, thereby leading to diseases. I think it is interesting to dig into these kinds of questions to dissect the underlying mechanisms for such enhancers leading to diseases.
- Figures are low-quality and hard to see. For example, rows in Fig. 6a are not readable.
- The authors mention in Discussion that "Genetic control of these trans regulatory mechanisms would likely heavily influence canonical gene expression downstream." However, they also mention in the paper that eRNAs are likely not playing a role in loop formation between enhancer and genes. These two statements are in contrary, as it is not clear why the eRNA that follow the trans model by recruiting chromatin modifiers and TFs can affect their disease-associated genes if there is no E-P contact.

-

(Remarks on code availability)

Reviewer #2

(Remarks to the Author)

Review of NCOMMS-24-34622 by Betti et al. (Gamazon)

“Genetically regulated enhancer RNA expression predicts enhancer- promoter contact frequency and reveals genetic mechanisms at complex trait-associated loci”

Summary

Betti et al. tackle interesting, pertinent, and difficult questions: how does eRNA transcription affect downstream gene expression, and is the eRNA landscape predictive for schizophrenia? The authors use a panel of analytical techniques that include machine learning to better correlate eRNA expression and canonical gene expression (derived from TWAS) to various chromatin features and ultimately downstream phenotypes. They have concluded that in trans mechanisms may be more prevalent in this context. These analytical techniques were then put to the test to see if there was predictive power in correlating eRNA to a very complicated phenotype: SCZ in UK biobank patients. Analysis was done on a combination of previously generated data by the lab and from publicly available data sets. The authors have made a marked improvement in predicting both gene expression and SCZ by analyzing eRNA expression over previous computational methods. They have also improved upon limitations of GWAS studies by including eRNA and TWAS.

Review

This study tests the hypothesis that like eQTLs, eRNAs might also be under some degree of genetic control, influencing human complex traits, like disease risk. While this study has the merits listed above, it falls short due to a few, but major fundamental issues listed below. Overall, this study is interesting and a good step forward for eRNA functional work, but would need to address the major concerns to be suitable for publication.

Major Comments:

- The definition of “1D eRNA” sounds like annotations for lncRNAs. As mentioned, these have different features than most eRNAs and may have to be considered separately.
- TWAS studies (including the ones you cited) are done from RNA-seq experiments. These would not properly capture eRNA transcription accurately. Instead, training sets should be done using direct assays of transcription (some sort of nuclear run on). If not possible, at least a single test set should be done on data generated from an assay that can directly measure transcription to confirm neural networks.
- For your training sets, it seems that R2 values were used as the primary selection metric. Have you considered using accuracy (i.e. accuracy in predicting level enh-gene contact) as a metric? If yes, please provide graphs.
- There seems to be a misunderstanding of in cis and in trans effects. ATAC signal and histone modifications may occur in both scenarios and is not a way to distinguish between the two. In general, cis has been used to describe eRNA acting within its own enhancer context (i.e. not soluble in the nucleus, where it can affect other enhancer/gene pairs). In contrast, trans has been used to define action outside its own gene context. This does not necessarily have to do with histone status or cohesin loops, as they may be relevant in both contexts. Therefore, methods used here to distinguish the two are insufficient.
- As there was no direct assays perturbing (increase or decrease) eRNA expression, it is hard to make any real mechanistic conclusions. How does modulation of eRNA affect the features tested (e.g., histone modifications, cohesin-complex recruitment, etc.)?

Minor Comments:

- Looping can occur without cohesin complex.
- Some representative browser tracks for data described in lines 283-305 would be useful.
- The authors used Kaiming initiation distributions in non-linear modeling of the neural network. While this seems to be a vast improvement compared to linear regression models, have the authors tried different non-linear models? Can they provide validation loss graphs for a few models?

(Remarks on code availability)

Version 1:

Reviewer comments:

Reviewer #1

(Remarks to the Author)

Thank you for addressing my comments. I have no further questions.

(Remarks on code availability)

Reviewer #2

(Remarks to the Author)

The authors have done a thorough job addressing my concerns, including additional data and revised text. In my original review, I expressed my opinion that the paper was unique, interesting, and addressed a challenging question, but fell short. I think the shortcomings have been addressed in the revised paper. This will be a nice contribution to the field.

(Remarks on code availability)

We thank the reviewers for their constructive feedback, which has substantially improved the manuscript. Below is a detailed response to their concerns and comments.

Reviewer 1

- How do you determine the number of enhancer-enhancer and enhancer-promoter interaction using Hi-C data? There are many ways to do so, such as using a statistical method like HiCDCPlus, which also considers distance as a feature, and thresholding based on false discovery rate to get only the statistically significant interactions. It is not clear how the authors come up with the E-E and E-P interactions.

We appreciate the reviewer asking for this important clarification. We have updated the text of the Methods section to a greater level of detail regarding Hi-C processing (lines 799-822):

“High-resolution Hi-C datasets that had been generated in the K562 leukemic cell line (accession number 4DNFI18UHVRO) and astrocytes of the cerebellum (accession number 4DNFIWCAQUIK) were retrieved from the 4D Nucleome¹⁹ Data Portal (<https://data.4dnucleome.org>) in mcool format. Raw sequencing reads underwent initial processing, contact matrix aggregation, and normalization using the gold standard Hi-C processing pipeline detailed at https://data.4dnucleome.org/resources/data-analysis/hi_c-processing-pipeline. The Hi-C dataset representing whole blood (K562) included 907,136,828 filtered reads, while the cerebellum-based dataset included 428,475,763. Both datasets showed similar quality control metrics, such as cis/trans ratio, % long-range intrachromosomal reads, and very good convergence (Supplementary Table 16). Contacts were normalized using the ICE (iterative correction and eigenvalue decomposition) algorithm¹⁰¹.

With the processed mcool file, Cooler (v0.8.2)¹⁰² was utilized to export contacts at a 10 kb resolution using the *cooler dump* function, and an R script was written to convert these outputs to BEDPE format (see GitHub). We defined an initial set of contact regions as those with one or more normalized contact in each Hi-C dataset. We then filtered these contact pairs down to a subset in which the respective 10 kb contact regions overlapped with either two annotated eRNAs or an eRNA and canonical gene. The eRNA annotations used were from the same collection utilized by the authors of the Human enhancer RNA Atlas (HeRA)⁹¹, and consisted of human eRNAs annotated by the ENSEMBL⁹⁷, FANTOM5⁹⁸, and Roadmap⁹⁵ consortia. Canonical gene annotations were obtained from GENCODE (v32)^{99,100}.”

- The author mention that in the case of cross-tissue generalization, the models trained on the whole blood cannot generalize to cerebellum data, but the models trained on the cerebellum data can generalize relatively well to the whole blood data. This also brings up a question about normalization of the data, as if the normalization is not done in a rigorous way, this issue can be seen. For example, the authors must make sure if their input data (eRNA expression) and output (E-E and E-P) have been normalized in a way that considers the library size and sequencing

depth of all these assays. Hi-C data in particular could end up with inconsistent numbers of called loops (E-E or E-P) if they have very different sequencing depth.

We thank the reviewer for bringing up this important question of data normalization and quality control for expression and Hi-C data, which could significantly impact downstream results if there are inconsistencies between datasets. The processed eRNA expression datasets were published as part of the Human enhancer RNA Atlas (HeRA). These data were processed using the gold standard RNA-seq pipeline developed by the GTEx Consortium. As part of this pipeline, all reads were normalized using the reads per million method.

All Hi-C data were processed using the gold standard 4D Nucleome pipeline detailed at https://data.4dnucleome.org/resources/data-analysis/hi_c-processing-pipeline. The Hi-C dataset representing whole blood (K562) utilized 907,136,828 filtered reads, while the cerebellum-based dataset consisted of 428,475,763. Both datasets showed similar quality control metrics, such as cis/trans ratio, % long-range intrachromosomal reads, and very good convergence. The ICE (iterative correction and eigenvalue decomposition) algorithm was used for normalization. These QC metrics have been added to Supplementary Table 16.

- The authors show low negative correlation between the predicted contact frequency and genomic distance in Fig S4. I am also interested to see the same plot for the true contact frequency vs genomic distance.

We appreciate the reviewer's suggestion to show the correlation between true contact frequency and genomic distance, in addition to the existing plot depicting predicted contact frequency versus genomic distance. These plots have been generated and added as additional panels for Supplementary Figure 9 (previously Supplementary Figure 4). Similar to the relationship between predicted contact frequency and genomic distance (Pearson correlation $R = -0.05$ in whole blood and $R = -0.08$ in cerebellum), we also observed a low negative correlation between true contact frequency and genomic distance (Pearson correlation $R = -0.10$ in whole blood and $R = -0.11$ in cerebellum).

Supplementary Figure 9. Correlation of deep learning model predictions with linear genomic distance. A weak negative correlation between genomic distance and predicted contact frequency was observed for both **a**, the model trained on GReX and Hi-C data from whole blood and **b**, the cerebellum-trained model. We also observed a weak negative correlation between genomic distance and observed contact frequency for both **c**, the model trained on GReX and Hi-C data from whole blood and **d**, the cerebellum-trained model.

- The authors mention “the mechanisms of eRNAs and genes in the context of SCZ risk are largely independent.” Do you have any insights on how the causal enhancers play a role in disease? It is great that the authors explore some mechanisms by which these causal eRNAs work, and they find that the trans model is dominant, meaning that eRNAs help making the chromatin more accessible rather than playing a role on loop formation. However, it is less clear how such enhancers lead to a disease. For example, in the context of a gene regulatory network, these enhancers might affect a cis gene/TF that might consequently affect their target genes, thereby leading to diseases. I think it is interesting to dig into these kinds of questions to dissect the underlying mechanisms for such enhancers leading to diseases.

We appreciate the reviewer raising this important concern regarding the mechanisms through which enhancers lead to disease and agree that dissecting the

underlying mechanisms of identified eRNAs would strengthen the overall impact of this study. We have added enhancer perturbation data from the K562 cell line for 109 complex-trait associated eRNAs. Of these 109 enhancers, we identified 14 (12.84%) that, when perturbed, resulted in a significant decrease in canonical gene expression. Among these eRNAs linked to canonical gene expression are transcripts associated with schizophrenia, mania, and hypertension. Notably, we did not observe Hi-C contacts between any of these 14 eRNAs and their target gene(s). We also find almost no SNP eQTLs shared across these eRNAs and their linked genes, suggesting that these eRNAs mediate canonical gene expression independently of chromatin contacts or shared SNP eQTLs. These results are described in lines 495-530:

“Enhancer perturbation links disease-associated eRNAs to canonical target gene expression

Epigenomic analysis of causal SCZ-associated eRNAs supported a context-independent, rather than contact-dependent model. These data alone, however, cannot fully explain the underlying mechanisms by which eRNA expression influences disease risk. To investigate whether disease-associated eRNAs have an effect on canonical gene expression, we utilized CRISPR perturbation assays in the K562 cell line targeting 109 complex trait-associated, transcribed enhancers identified by TWAS (Supplementary Tables 12-13). Of these 109 enhancers, we identified 14 (12.84%) for which CRISPR perturbation resulted in a significant change in expression of a corresponding gene. Notably, we did not observe Hi-C contacts between any of these 14 eRNAs and their target gene(s). We also performed eQTL mapping in whole blood for both eRNAs and canonical genes. Among the 22 unique eRNA-gene pairs identified using CRISPR perturbation, we observed only one mapped SNP eQTL (FDR < 0.1) that was shared by both the eRNA and canonical gene in a pair. Thus, these eRNAs appear to mediate canonical gene expression independently of chromatin contacts or shared SNP eQTLs.

Among some of the disease-relevant eRNAs linked to canonical gene expression were ENSR00000013481 and ENSR00000032851, both associated with SCZ; ENSR00000117322, associated with manifestations of mania or irritability; and ENSR00000320642, associated with hypertension. Perturbation of SCZ-associated eRNAs ENSR00000013481 and ENSR00000032851 resulted in decreased expression of *VPS45* (fold change = 0.75, $p = 1.21 \times 10^{-3}$) and *NT5C2* (fold change = 0.74, $p = 7.66 \times 10^{-4}$), respectively. Both genes have previously been implicated in SCZ and reach significance in our canonical gene-based TWAS of SCZ⁵⁵⁻⁵⁷.

In addition to the two SCZ-associated eRNAs, perturbation of ENSR00000117322 (associated with Manifestations of mania or irritability) resulted in decreased expression of the SCZ-associated gene *RTN4* (fold change = 0.83, $p = 6.91 \times 10^{-9}$), while perturbation of hypertension-associated eRNA ENSR00000320642 resulted in decreased *MRPS10* expression (fold change = 0.82, $p = 1.42 \times 10^{-8}$). This gene codes for a mitochondrial ribosomal protein and has previously been linked to cardiac disorders⁵⁸⁻⁶⁰. Neither of these eRNA-linked genes reached significance in a canonical gene-based TWAS.”

- Figures are low-quality and hard to see. For example, rows in Fig. 6a are not readable.

We thank the reviewer for indicating that some of the figures were difficult to read. When exporting this revised manuscript, we have verified that none of the embedded figures were being compressed as they were previously. All raw figures are exported at a resolution of 300 dpi or greater. Figure 6a has also been modified to depict the top 50 eRNA-based TWAS associations, rather than the top 100 that were originally shown, to make the text more readable. We have also increased both the font size and amount of space between the the phenotype labels to improve legibility.

Figure 6. eRNA eQTLs are associated with complex traits across the phenome and help to explain 63% more GWAS signals than canonical gene eQTLs alone. **a**, Depicted are the top 50 UK Biobank traits ranked by number of significant ($p < 2.60 \times 10^{-10}$) eRNA-tissue associations. The color for a trait denotes the p-value for the most significant association with the trait. **b**, We identified 18,815 genome-wide significant ($p < 5 \times 10^{-8}$) GWAS signals within the UK Biobank that only colocalized with an eRNA eQTL, representing a 63% increase over the number of independent associations that can be explained by canonical gene eQTLs alone.

- The authors mention in Discussion that “Genetic control of these trans regulatory mechanisms would likely heavily influence canonical gene expression downstream.” However, they also mention in the paper that eRNAs are likely not playing a role in loop formation between enhancer and genes. These two statements are in contrary, as it is not clear why the eRNA that follow the trans model by recruiting chromatin modifiers and TFs can affect their disease-associated genes if there is no E-P contact.

We appreciate the reviewer's thoughtful comment and acknowledge the need to clarify the mechanisms by which eRNAs can influence canonical gene expression without being involved in enhancer-promoter contact formation. In our response to point four from the reviewer, we utilized enhancer CRISPR perturbation data to what, if any, effect perturbation of a disease-associated eRNA would have on canonical gene expression. Of the 109 complex trait-associated eRNAs that were perturbed, we identified a subset of 14 (~12%) that resulted in a significant decrease in gene expression, despite none of them showing evidence of chromatin interactions in Hi-C data.

A complete global dissection of the epigenetic mechanisms of eRNA activity, while incredibly interesting, would require an extensive set of epigenomic assays beyond the scope of this study. However, we believe that the enhancer perturbation data that we now include strengthens our assertion that disease-associated eRNAs influence canonical gene expression without being involved in enhancer-promoter contact formation. We have updated the discussion to reference these new results and discuss potential mechanisms through which this contact-free gene regulation may occur.

Reviewer 2

- The definition of "1D eRNA" sounds like annotations for lncRNAs. As mentioned, these have different features than most eRNAs and may have to be considered separately.

We thank the reviewer for pointing out this important distinction between 2D eRNAs and 1D eRNAs and that there is a high degree of overlap between lncRNAs and 1D eRNAs. In the manuscript, we note that that the previously published eRNA expression dataset that we utilize is derived from GTEx RNA-seq data, which is enriched for polyadenylated transcripts. Thus, it is possible that 2D eRNAs are underrepresented in this dataset.

In the results section, we now characterize the proportion of eRNAs from our original training dataset that fit the definition of a 2D eRNA versus a 1D eRNA. Previous studies have described 2D eRNAs as having a length below 2 kb, while 1D eRNAs should be longer than 2 kb. Of the 14,471 unique eRNA transcripts included in our trained models, 13,580 (93.84%) were less than 2 kb in length, while only 891 (6.16%) has a length greater than or equal to 2 kb (Supplementary Table 1). The median eRNA transcript length was 550 bp (Supplementary Figure 1). These results suggest that despite transcripts in the training set undergoing selection for polyadenylated transcripts, the vast majority still exhibit characteristics typical of 2D eRNAs. These new results are described in lines 88-99 of the Results section:

"Because eRNAs were quantified from RNA-seq data, which underwent selection for polyadenylated transcripts prior to sequencing⁹, we realized that 2D eRNAs, which are not polyadenylated, might be underrepresented in the training dataset. Previous

characterizations of eRNAs have described 2D transcripts as having a length below 2 kb, while 1D eRNAs are longer than 2 kb^{15–17}. Using transcript length, we characterized the proportion of eRNAs in our training dataset that fit the definition of a 2D eRNA versus a 1D eRNA. Of the 14,471 unique eRNA transcripts included in our trained models, 13,580 (93.84%) were less than 2 kb in length, while only 891 (6.16%) has a length greater than or equal to 2 kb (Supplementary Table 1). The median eRNA transcript length was 550 bp (Supplementary Figure 1). These results suggest that despite transcripts in the training set undergoing selection for polyadenylated transcripts, the vast majority still exhibit characteristics typical of 2D eRNAs.”

Supplementary Figure 1. Distribution of transcript length among the 14,471 unique eRNAs included in the GReX models.

- TWAS studies (including the ones you cited) are done from RNA-seq experiments. These would not properly capture eRNA transcription accurately. Instead, training sets should be done using direct assays of transcription (some sort of nuclear run on). If not possible, at least a single test set should be done on data generated from an assay that can directly measure transcription to confirm neural networks.

We appreciate the reviewer making this important point that RNA-seq experiments, on which ours and previous eRNA-based TWAS studies have been based, do not capture eRNA transcription as well as a nuclear run on experiment would. While the primary focus of this study is genetically regulated expression (GRex) of eRNAs, which is predicted using the TWAS models we present, we have trained additional contact models using direct assays of transcription. Using GRO-cap data and Hi-C from the K562 cell line, we trained a neural network-based model to predict chromatin contact frequency between an enhancer-enhancer or enhancer-gene pair, given measured transcription from the GRO-cap assay. This model showed comparable performance with our contact model trained on GRex. We also tested our GRex-based contact model on the nuclear run on dataset and found that it retained predictive performance in the GRO-cap dataset ($R^2 = 0.15$). These results are described in lines 158-177 in the manuscript, and the optimal model architecture and training curves for the nuclear run-on-based model are depicted in Supplementary Figure 7:

“Training a baseline neural network-based contact model using directly assayed eRNA and gene expression

Prior to training a neural network using GRex, we trained an initial baseline contact prediction model using eRNA and gene transcript counts quantified via a nuclear run-on assay in the K562 cell line (Supplementary Figure 6). As nuclear run-on assays are considered a gold standard for detecting nascent eRNA transcription, a model trained on these expression data should be an ideal benchmark against which a GRex-based model can be compared.

An initial neural network model was trained using eRNA and canonical gene transcription to predict chromatin contacts in the K562 cell line. Hyperparameter tuning was conducted across a pre-defined search space, and five-fold cross validation was used to quantify model performance (see **Methods**). The optimal model architecture (Supplementary Figure 7a) consisted of two neurons in the input layer (for the normalized expression levels of the upstream and downstream transcripts), two hidden layers with 150 neurons each (using a ReLU activation function), and a single output neuron. Hidden weights were initialized using a normal distribution, while those in the output neuron were initialized with zeros. The model was trained over 90 epochs using a batch size of 160. The Adagrad²⁰ optimizer was utilized with a learning rate of 0.2. This optimal model architecture achieved a mean R^2 of 0.23 across the validation folds (Supplementary Figure 7b) and 0.27 in the independent test set.”

a**b**
Supplementary Figure 7. Nuclear run-on-based baseline model. a, Grid search across 13 hyperparameters was used to achieve the optimal model architecture (see **Methods**). **b**, The neural network regressor was trained for 90 epochs, achieving a mean prediction R^2 of 0.23 across the validation folds and 0.27 in the independent test set.

- For your training sets, it seems that R^2 values were used as the primary selection metric. Have you considered using accuracy (i.e. accuracy in predicting level enh-gene contact) as a metric? If yes, please provide graphs.

We thank the reviewer for their suggestion to consider using accuracy as a selection metric during training of the neural network-based contact model, as R^2 is just one of multiple different metrics we could have chosen. To test whether the optimal model hyperparameters or performance of the fully trained model greatly differ based on epoch scoring metric, we re-trained contact models from scratch using accuracy metric root mean square error (RMSE) rather than R^2 . While the final combination of optimal hyperparameters was slightly different from the models that used R^2 , we found validation performance to be comparable between the two sets of models (Whole blood: $R^2 = 0.22$ and RMSE = 61.23 in the model trained using R^2 versus R^2

= 0.20 and RMSE = 61.80 in the model trained using RMSE; Cerebellum: $R^2 = 0.37$ and RMSE = 32.48 in the model trained using R^2 versus $R^2 = 0.39$ and RMSE = 27.97 in the model trained using RMSE.). The training curves of these RMSE-based models are shown in Supplementary Figure 12.

Supplementary Figure 12. Training neural GReX-based neural network models using root mean squared error (RMSE) as the selection criterion. a, The optimal model for whole blood consisted of two hidden layers, each with a size of 130. The ReLU6 activation function was used in the hidden layers. Weights in the hidden layers were initialized using a Kaiming uniform distribution, while those in the output layer were initialized with zeros. **b,** The model was trained for 50 epochs using a batch size of 60. The NAdam optimizer was implemented with a learning rate of 0.002. The optimal model achieved a RMSE of 61.80 in the validation set, which corresponds with an R^2 of 0.20. The model that used R^2 as the selection criterion achieved a comparable validation R^2 of 0.22. **c,** The optimal model for cerebellum consisted of two hidden layers, each with a size of 90. The Softsign activation function was used in the hidden layers. Weights in the hidden layers were initialized using a normal distribution, while those in the output layer were initialized with zeros. **d,** The model was trained for 80 epochs using a batch size of 120. The Adagrad optimizer was implemented with a learning rate of 0.2. The optimal model achieved a RMSE of 25.47 in the validation set, which corresponds with an R^2 of 0.39. The model that used R^2 as the selection criterion achieved a comparable validation R^2 of 0.37.

- There seems to be a misunderstanding of in cis and in trans effects. ATAC signal and histone modifications may occur in both scenarios and is not a way to distinguish between the two. In general, cis has been used to describe eRNA acting within its own enhancer context (i.e. not soluble in the nucleus, where it can affect other enhancer/gene pairs). In contrast, trans has been used to define action outside its own gene context. This does not necessarily have to do with histone status or cohesin loops, as they may be relevant in both contexts. Therefore, methods used here to distinguish the two are insufficient.

We appreciate the reviewer's thoughtful comment and acknowledge the need for clarification regarding our description of *cis* versus *trans* mechanisms. To avoid confusion, we have modified our naming to "contact-dependent" and "contact independent" mechanisms. In addition to these epigenomic assays, we have also since added enhancer perturbation results for 109 complex trait-associated eRNAs in the K562 cell line.

- As there was no direct assays perturbing (increase or decrease) eRNA expression, it is hard to make any real mechanistic conclusions. How does modulation of eRNA affect the features tested (e.g., histone modifications, cohesin-complex recruitment, etc.)?

We thank the reviewer for raising the need for perturbation assays to better understand the underlying mechanisms of how eRNA expression influences disease biology. We now include an experimental dataset in the K562 cell line, in which 109 complex trait-associated enhancers were perturbed using CRISPRi. Of these 109, we identified 14 eRNAs that, when perturbed, resulted in significantly decreased gene expression, despite the enhancer-gene pair not being in physical contact (based on Hi-C data from the same cell line) and them sharing almost no SNP eQTLs. These results are described in lines 495-530:

“Enhancer perturbation links disease-associated eRNAs to canonical target gene expression

Epigenomic analysis of causal SCZ-associated eRNAs supported a context-independent, rather than contact-dependent model. These data alone, however, cannot fully explain the underlying mechanisms by which eRNA expression influences disease risk. To investigate whether disease-associated eRNAs have an effect on canonical gene expression, we utilized CRISPR perturbation assays in the K562 cell line targeting 109 complex trait-associated, transcribed enhancers identified by TWAS (Supplementary Tables 12-13). Of these 109 enhancers, we identified 14 (12.84%) for which CRISPR perturbation resulted in a significant change in expression of a corresponding gene. Notably, we did not observe Hi-C contacts between any of these 14 eRNAs and their target gene(s). We also performed eQTL mapping in whole blood for both eRNAs and canonical genes. Among the 22 unique eRNA-gene pairs identified using CRISPR perturbation, we observed only one mapped SNP eQTL (FDR < 0.1) that was shared by both the eRNA and canonical gene in a pair. Thus, these eRNAs appear to mediate canonical gene expression independently of chromatin contacts or shared SNP eQTLs.

Among some of the disease-relevant eRNAs linked to canonical gene expression were ENSR00000013481 and ENSR00000032851, both associated with SCZ; ENSR00000117322, associated with manifestations of mania or irritability; and ENSR00000320642, associated with hypertension. Perturbation of SCZ-associated eRNAs ENSR00000013481 and ENSR00000032851 resulted in decreased expression of *VPS45* (fold change = 0.75, $p = 1.21 \times 10^{-3}$) and *NT5C2* (fold change = 0.74, $p = 7.66 \times 10^{-4}$), respectively. Both genes have previously been implicated in SCZ and reach significance in our canonical gene-based TWAS of SCZ⁵⁵⁻⁵⁷.

In addition to the two SCZ-associated eRNAs, perturbation of ENSR00000117322 (associated with Manifestations of mania or irritability) resulted in decreased expression of the SCZ-associated gene *RTN4* (fold change = 0.83, $p = 6.91 \times 10^{-9}$), while perturbation of hypertension-associated eRNA ENSR00000320642 resulted in decreased *MRPS10* expression (fold change = 0.82, $p = 1.42 \times 10^{-8}$). This gene codes for a mitochondrial ribosomal protein and has previously been linked to cardiac disorders⁵⁸⁻⁶⁰. Neither of these eRNA-linked genes reached significance in a canonical gene-based TWAS.”

Minor Comments:

- Looping can occur without cohesin complex.

We appreciate the reviewer pointing out that although the cohesin complex is involved in chromatin loop formation, looping can occur without cohesin. This point has been clarified in the manuscript text.

- Some representative browser tracks for data described in lines 283-305 would be useful.

We thank the reviewer for their suggestion to visually represent some examples of the data described in this section as browser tracks. These visualizations have been added to the manuscript as Supplementary Figure 10.

a**b**
Supplementary Figure 10. Epigenomic patterns inform possible mechanisms by which causal eRNAs influence risk for SCZ. **a**, If causal eRNAs influence SCZ risk via direct mediation of enhancer-gene interactions, we would expect to observe a strong enrichment of either RAD21 and SMC3 or CTCF. However, only a small proportion of eRNAs were enriched for these features, and those that were showed no evidence of contact with a causal gene. **b**, If causal eRNA expression plays a role in maintaining an open chromatin state, we should expect to observe an enrichment of ATAC-seq and DNase-seq peaks, as well as EP300 and H3K27ac, in addition to a depletion of H3K27me3. We observed enrichment for at least one of these marks in 74% of causal SCZ-associated eRNAs in the brain.

- The authors used Kaiming initiation distributions in non-linear modeling of the neural network. While this seems to be a vast improvement compared to linear regression models, have the authors tried different non-linear models? Can they provide validation loss graphs for a few models?

We appreciate the reviewer's suggestion to test how other non-linear models perform at this contact frequency prediction task. In addition to the neural network and linear regression that were initially presented, we now also provide results and loss curves for the following four non-linear models: polynomial regression, random forest regression, support vector regression, and gradient boosting regression. Of these non-linear models that were tested, gradient boosting regression performed the best in whole blood ($R^2 = 0.06$), while random forest regression performed best in cerebellum ($R^2 = 0.13$). Nevertheless, performance of each respective model was substantially lower than that of the neural network trained using the same data ($R^2 = 0.23$ and $R^2 = 0.37$ in the respective validation sets for whole blood and cerebellum). We have added the corresponding loss and graphs as Supplementary Figures 4 and 5.

Supplementary Figure 4. Training curves for non-linear models in whole blood. **a**, We fit a series of polynomial regression models, testing a range of values for degree (1-10). The best performing model ($R^2 = 0.02$) used a degree of 7. **b**, We fit a series of random forest regression models, testing different numbers of trees (10-100). The best performing model ($R^2 = 0.04$) used 40 trees. **c**, We fit a series of support vector regression models, testing different values of epsilon (0.1-5.0). The best performing model ($R^2 = 0.00$) used an epsilon value of 5. **d**, We fit a series of gradient boosting regression models, testing different numbers of boosting stages (10-300). The best performing model ($R^2 = 0.08$) used 290 boosting stages.

Supplementary Figure 5. Training curves for non-linear models in cerebellum. *a*, We fit a series of polynomial regression models, testing a range of values for degree (1-10). The best performing model ($R^2 = 0.08$) used a degree of 8. *b*, We fit a series of random forest regression models, testing different numbers of trees (10-100). The best performing model ($R^2 = 0.13$) used 100 trees. *c*, We fit a series of support vector regression models, testing different values of epsilon (0.1-5.0). The best performing model ($R^2 = 0.01$) used an epsilon value of 5. *d*, We fit a series of gradient boosting regression models, testing different numbers of boosting stages (10-300). The best performing model ($R^2 = 0.13$) used 290 boosting stages.